# AdaBoN: Adaptive Best-of-$N$ Alignment

## Abstract

Recent advances in test-time alignment methods, such as Best-of-$N$ sampling, offer a simple and effective way to steer language models (LMs) toward preferred behaviors using reward models (RM). However, these approaches can be computationally expensive, especially when applied uniformly across prompts without accounting for differences in alignment difficulty. In this work, we propose a prompt-adaptive strategy for Best-of-$N$ alignment that allocates inference-time compute more efficiently. Motivated by latency concerns, we develop a two-stage algorithm: an initial exploratory phase estimates the reward distribution for each prompt using a small exploration budget, and a second stage adaptively allocates the remaining budget using these estimates. Our method is simple, practical, and compatible with any LM-RM combination. Empirical results on prompts from the AlpacaEval, HH-RLHF, and PKU-SafeRLHF datasets for 12 LM–RM pairs and 50 different batches of prompts show that our adaptive strategy outperforms the uniform allocation with the same inference budget. Moreover, we show that our adaptive strategy remains competitive against uniform allocations with $20\%$ larger inference budgets and improves in performance as the batch size grows.

## 1 Introduction

Language Models (LMs) have demonstrated human-like capabilities across a wide range of tasks, including mathematics, coding, and creative writing (Brown et al., 2020; Achiam et al., 2023). While pre-training on massive corpora equips these models with extensive knowledge, it is crucial that their responses at inference-time adhere to ethical standards and safety guidelines. A common approach involves leveraging preference data to steer the model toward more desirable outputs. For example, post-training methods such as Reinforcement Learning with Human Feedback (RLHF) (Christiano et al., 2017; Ouyang et al., 2022), Direct Preference Optimization (DPO) (Rafailov et al., 2023), and its variants (Glaese et al., 2022), fine-tune the model weights, while constraining the updated model to remain close to a reference model.

Despite its empirical success, post-training methods are computationally expensive and can introduce unintended and opaque changes to the base model (Ouyang et al., 2022; Bai et al., 2022). **Inference-time alignment** techniques leave the model weights untouched, but modify the *decoding strategy* to guide the output distribution at inference time (Li et al., 2023a; Wang et al., 2024a; 2025).

One of the simplest and most popular inference-time alignment methods is **Best-of-$N$** sampling, which has gained significant traction due to its simplicity, model-agnostic nature, and strong empirical performance (Nakano et al., 2021). Given a prompt and a reward model that scores outputs by alignment quality, Best-of-$N$ sampling generates $N$ responses from the base LM and returns the one with the highest reward. Despite its simplicity, Best-of-$N$ alignment remains competitive with fine-tuning approaches like DPO and RLHF. Its transparent mechanics makes it amenable to theoretical analyses (Gui et al., 2024; Beirami et al., 2024; Huang et al., 2025; Yang et al., 2024), efficiency improvements (Qiu et al., 2024; Sun et al., 2024; Wang et al., 2025), and use for synthetic data generation in down-stream fine-tuning tasks (Touvron et al., 2023; Dubois et al., 2023)

Yet, a key limitation of Best-of-$N$ sampling is its **lack of adaptivity**. In practice, the value of $N$ is typically chosen via hyperparameter tuning and applied uniformly across all prompts, regardless of their difficulty (Nakano et al., 2021). This can be inefficient: some prompts may require only a few samples to yield a high-reward response, while others may benefit from more extensive sampling

(Damani et al., 2024). Since one might need to pick $N$ as large as $10,000$ to be competitive with post-training methods (Gao et al., 2023), a naive uniform allocation leads to wasted computation.

In light of this issue, we introduce a **prompt-adaptive** approach to Best-of-$N$ alignment by building on recent progress in input-adaptive compute allocation (Snell et al., 2024; Damani et al., 2024). Specifically, we consider a setting in which we are given a batch of prompts $x_1, \ldots, x_K$ and a per-prompt inference budget $B$, measured in the number of forward passes or queries to the LM. Our goal is to allocate the total budget $BK$ across the prompts to maximize the cumulative reward obtained via Best-of-$N$ sampling, where $N$ may now vary across prompts. *We focus on the regime where the batch size $K$ is small and the per-prompt budget $B$ is large.* This is relevant for personalized on-device inference, where models are small and hence compute per prompt is large, while the number of prompts is limited (Zhang et al., 2024b). Our main contributions are as follows.

(1) We find that the per-prompt reward distributions for the LM-RM pairs we consider are *smooth and easy to learn*.

(2) Leveraging this, we propose a simple yet effective *two-stage* Adaptive Best-of-$N$ (AdaBoN) allocation scheme. In the first-stage, we use a small exploration budget to estimate reward distributions for each prompt. In the second-stage, we use these estimates to compute the marginal value of allocating additional samples and apply a greedy algorithm to assign the remaining budget accordingly.

(3) We define two new evaluation metrics, termed the Batch Win Rate (BWR) and Expected Survival Time (EST), which measure the ability for AdaBoN to outperform the uniform allocation and compete against larger inference budgets respectively.

(4) Using these metrics, we evaluate AdaBoN on prompts from the AlpacaEval, HH-RLHF, and PKU-SafeRLHF datasets. We sample 50 batches of prompts and find that:

    a. AdaBoN consistently outperforms the uniform allocation across the 50 batches, with some batches having win rates as high as $70\%$.

    b. AdaBoN is competitive against uniform allocations with $20\%$ *larger* inference budgets.

    c. AdaBoN improves in performance as the batch grows for the majority of LM-RM pairs and is robust to changes in the inference budget, continuing to obtain win rates significantly larger than $0.50$ for smaller and larger inference budgets.

    d. AdaBoN minimizes latency and has only a single hyperparameter that needs to be tuned. Even then, we find that a single choice of this hyperparameter works well across all experiments we run.

## 1.1 RELATED WORK

**Inference-time Alignment and Best-of-$N$ sampling.** Compared to fine-tuning based approaches, like DPO and RLHF, test-time alignment aims to steer a base policy purely at inference-time, without changing the model weights. Some popular inference-time alignment methods include Best-of-$N$ sampling Gao et al. (2023); Stiennon et al. (2022), majority voting (Wang et al., 2022), weighted majority voting (Li et al., 2023b), hypothesis re-weighting Lee et al. (2024), and Markov chain Monte Carlo (Faria and Smith, 2025). Controlled decoding (Mudgal et al., 2023) and ARGS (Khanov et al., 2024) also fall into this broader family of test-time alignment methods.

Of particular interest to us is Best-of-$N$ sampling, which has emerged as a prominent inference-time alignment strategy, offering a simple yet effective mechanism for aligning LM outputs with human preferences. Originally introduced as a baseline for inference-time alignment (Nakano et al., 2021), Best-of-$N$ has since found widespread use, both as a standalone method and as part of larger alignment pipelines (Touvron et al., 2023). In addition to its standalone appeal, Best-of-$N$ has been integrated into more complex frameworks such as rejection sampling variants of DPO (Liu et al., 2023) and RLHF (Dong et al., 2023).

A key aspect of Best-of-$N$'s empirical success is its compelling reward-KL tradeoff curves (Gao et al., 2023; Mudgal et al., 2023; Eisenstein et al., 2023). In particular, compared to KL-regularized reinforcement learning (RL) techniques, Best-of-$N$ often achieves comparable rewards while staying closer to the base model's distribution. This empirical behavior has been substantiated theoretically, with several works deriving tight estimates of the KL divergence between the Best-of-$N$ policy

and the base policy (Coste et al., 2023; Gao et al., 2023; Go et al., 2023; Beirami et al., 2024; Gui et al., 2024). Yang et al. (2024) prove that the Best-of-$N$ and KL-regularized RL policies converge to the same asymptotic behavior under reasonable assumptions.

Recent work has also proposed making Best-of-$N$ more efficient through speculative decoding, such as speculative rejection sampling (Sun et al., 2024) and TreeBoN (Qiu et al., 2024), which prune low-reward candidates early. While these methods reduce the per-prompt cost of BoN, they do not address the problem of distributing a fixed query budget across multiple prompts.

**Input-adaptive Inference Allocation.** The most closely related work to us is by Damani et al. (2024), who address the same inference budget allocation problem: given a batch of prompts $x_1, \ldots, x_K$, the goal is to distribute a total budget across them to maximize the cumulative maximum per-prompt reward. While the setup is similar, their approach differs from ours in three ways.

First, their method relies on training an auxiliary model that predicts the expected marginal gain in reward from allocating additional queries to a prompt. At test time, this model is queried for each prompt in the batch to obtain a vector of estimated gains. This batch of vectors of estimated gains is then used to determine the final allocation. A key limitation of this strategy is that the auxiliary model must be retrained whenever the domain, underlying LM, decoding strategy, or total inference budget changes. This makes it less flexible and potentially expensive, especially for large inference budgets. In contrast, our method is entirely at test-time and hence model-agnostic: it requires no auxiliary training, works out-of-the-box for any LM-RM pair, and adapts to the inference budget.

Second, their focus is on the regime where the batch size is large and the per-prompt budget is small. We consider the opposite setting: small batch sizes with large per-prompt budgets. This is particularly relevant for on-device LMs, which are smaller and cheaper to query, making high per-prompt budgets more feasible. In this regime, our approach benefits from directly estimating marginal gains via Monte Carlo sampling, removing the need for an auxiliary model. In contrast Damani et al. (2024)'s method does not observe significant improvements for large inference budgets.

Third, while our work targets alignment with real-valued reward models, much of Damani et al. (2024)'s evaluation focuses on binary rewards in domains such as math and coding. Although they do include a real-valued reward setting in the chat domain, their experiments are limited to a single LM, a single RM, and a single batch of prompts. In contrast, we conduct a broad empirical study covering 12 LM–RM pairs and 50 distinct batches, providing a more comprehensive assessment of prompt-adaptive alignment. In addition to Damani et al. (2024), there are few other works relevant to us. We provide a summary of them in Appendix B.

## 2 PRELIMINARIES

### 2.1 NOTATION

Let $\mathcal{X}$ denote the space of prompts and $\mathcal{Y}$ be the space of responses. A LM $\pi : \mathcal{X} \to \Delta\mathcal{Y}$ maps a prompt to a distribution over responses, where we let $\Delta\mathcal{Y}$ denote the set of all distributions on $\mathcal{Y}$. A reward model is a function $r : \mathcal{X} \times \mathcal{Y} \to \mathbb{R}$ that maps a prompt and a response to a real-value. Given a prompt $x \in \mathcal{X}$, LM $\pi$, and reward model $r$, we will use $r \circ \pi(x)$ to denote the distribution over rewards induced by passing $x$ to $\pi$, sampling $y \sim \pi(x)$, and then computing $r(x, y)$. Throughout the paper, we use $B$ to denote the *per-prompt* inference budget and $K$ to denote the number of prompts in a batch. Thus, for per-prompt budget $B$ and batch size $K$, the total budget is $BK$. Finally, we define $[B] := \{1, \ldots, B\}$ and abbreviate a sequence $z_1, \ldots z_n$ as $z_{1:n}$.

### 2.2 INFERENCE-TIME ALIGNMENT AND BEST-OF-$N$ SAMPLING

When aligning the responses of a LM with human values, one common approach is to use an external reward model $r : \mathcal{X} \times \mathcal{Y} \to \mathbb{R}$ to evaluate the quality of its responses. Usually, the reward model is trained using preference data and assigns higher scores to responses that exhibit desirable properties, e.g. like helpfulness, harmlessness, coherence, relevance, and fluidity. In *inference-time* alignment, the goal is modify the decoding procedure of $\pi$ so as to maximize the the reward model $r$. Perhaps the simplest way to do this is via Best-of-$N$ sampling, which has received significant interest due to being light-weight and model agnostic. Given a LM $\pi$, a sample budget $N \in \mathbb{N}$, a reward model $r$,

and a prompt $x$, the Best-of-$N$ alignment procedure involves sampling $N$ responses $y_1, \ldots, y_N \sim \pi(x)$ and returning $\arg\max_{y \in \{y_1, y_2, \ldots, y_N\}} r(x, y)$.

Despite its flexibility, Best-of-$N$ alignment suffers from high computational costs due to its lack of adaptivity – $N$ inference calls are made for every prompt $x \in \mathcal{X}$, where the $N$ is typically chosen via hyperparameter search and can be very large. This can often be wasteful if certain prompts are "easier" to generate aligned responses for than others. The focus of this work is to design a prompt-adaptive version of Best-of-$N$ alignment.

## 2.3 THE INFERENCE ALLOCATION PROBLEM

In this paper, we consider *adaptive* Best-of-$N$ alignment in the context of the following *resource allocation problem*. We are presented with a collection of $K$ prompts $x_{1:K}$ and a *per-prompt* inference budget $B$, measured in the total number of queries we can make to the base LM $\pi$. An allocation $a \in [BK]^K$, is a vector of size $K$ such that $\sum_{i=1}^{K} a_i \leq BK$. Here, $a_i$ represents the number of LM calls allocated to prompt $x_i$. For a fixed allocation $a \in [BK]^K$, the quantity

$$\mathbb{E}_{R_{i,j} \sim r \circ \pi(x_i)} \left[ \sum_{i=1}^{K} \max_{j=1,\ldots,a_i} R_{i,j} \right]. \tag{1}$$

is the cumulative expected reward obtained by running Best-of-$N$ sampling with base policy $\pi$ and reward model $r$. The goal is to find an allocation that maximizes Equation 1.

In general, without knowledge of the true distributions $r \circ \pi(x_1), \ldots, r \circ \pi(x_K)$, the uniform allocation is the minimax optimal *non-adaptive* allocation. By non-adaptive, we mean that the uniform allocation does not depend on the realization of some of the realized rewards. This is in contrast to an *adaptive* allocation, which may depend on *some* of the realized rewards.

Unsurprisingly, adaptivity is crucial for maximizing the cumulative sum of per-prompt rewards. As a simple example, consider the case where there are two prompts $x_1$ and $x_2$ and let $B = 25$ be the per-prompt budget. Suppose the reward distribution for $x_1$ and $x_2$ are Bernoulli distributions with parameters $p_1 = 0.95$ and $p_2 = 0.05$ respectively. The non-adaptive uniform allocation allocates 25 samples each to $x_1$ and $x_2$, resulting in an expected reward of $2 - (1 - p_1)^{25} - (1 - p_2)^{25}$. Alternatively, consider the following simple two-stage allocation procedure. For each prompt $x_1$ and $x_2$, sample $d = 10$ rewards. Let $R_{1:d}^1$ and $R_{1:d}^2$ be the realized rewards for prompt $x_1$ and $x_2$ respectively. Then, if $\max\{R_{1:d}^1\} = \max\{R_{1:d}^2\} = 1$, the procedure allocates the remaining $2B - 2d = 30$ queries arbitrarily. On the other hand, if $\max\{R_{1:d}^1\} = 1$ and $\max\{R_{1:d}^2\} = 0$, the procedure allocates the remaining $2B - 2d = 30$ queries to prompt $x_2$ and vice versa. Finally, if both $\max\{R_{1:d}^1\} = \max\{R_{1:d}^2\} = 0$, the procedure uniformly allocates the remaining $2B - 2d$ queries among $x_1$ and $x_2$. Brute force computation shows that the expected reward of the two-stage adaptive allocation is $1.87$ while the expected reward of the uniform allocation is only $1.72$.

While simple, the previous examples highlights the power of adaptivity for the inference allocation problem. This is in line with the results of Snell et al. (2024), who highlight the importance of estimating prompt "difficulty" for optimal test-time compute scaling. To that end, our focus in this paper will be towards designing *adaptive* allocation policies $\mathcal{A}$ which *sequentially* allocate the total budget $BK$ across the $K$ different prompts. The policy $\mathcal{A}$ need not allocate the inference budget all at once, but can allocate its budget one at a time, adapting to the past realized rewards. In this sense, the allocation returned by $\mathcal{A}$ is a random variable, where the randomness is due to the randomness of the base LM $\pi$ as well as any internal randomness that $\mathcal{A}$ decides to use.

Given an allocation policy $\mathcal{A}$ and a matrix of rewards $\{R_{i,j}\}_{i \in [K], j \in [BK]}$, where $R_{i,j} \sim r \circ \pi(x_i)$, we will use $\mathcal{A}(\{R_{i,j}\}, B)$ to denote the distribution over allocations induced by $\mathcal{A}$, when the realized rewards are $\{R_{i,j}\}_{i \in [K], j \in [BK]}$ and the per-prompt inference budget is $B$. Rather than choosing a fixed allocation, our objective now is to design a (randomized) allocation policy $\mathcal{A}$ so as to maximize

$$\mathbb{E}_{\substack{R_{i,j} \sim r \circ \pi(x_i) \\ A \sim \mathcal{A}(\{R_{i,j}\}, B)}} \left[ \sum_{i=1}^{K} \max_{j=1,\ldots,A_i} R_{i,j} \right].$$

Although the space of allocation policies is massive, in this paper, we will focus our attention on **two-stage** allocation policies $\mathcal{A}$. These are policies which use a pre-determined per-prompt initial

budget $d \leq B$ to *explore* each prompt, before *committing* to a fixed allocation for the remaining budget. Our focus on two-stage policies is motivated by **latency** concerns – as the adaptivity of $\mathcal{A}$ increases, one pays in latency as calls to the base LM $\pi$ can no longer be parallelized. This is a concern with existing adaptive Best-of-$N$ sampling methods (Manvi et al., 2024; Sun et al., 2024).

## 3  AN ADAPTIVE TWO-STAGE ALLOCATION POLICY

In this section, we present a lightweight, two-stage allocation policy for the inference allocation problem outlined in Section 2.3. Compared to Damani et al. (2024), our method does not require training of any auxiliary model and can be used in a black-box fashion for *any* LM-RM combination.

The two-stage allocation policy follows in three steps. In the first step, for each prompt $x_i$ in the batch, we sample $d \leq B$ times from $r \circ \pi(x_i)$ and construct an estimate $\hat{D}_i$ of $r \circ \pi(x_i)$ using a pre-specified distribution estimation procedure $f$. The total cost of this step is $dK$. In the second step, for each prompt $x_i$ in the batch, we use $\hat{D}_i$ to estimate the expected *gain* of sampling $j$ more times from $r \circ \pi(x_i)$, for $j = 1, \ldots, (B-d)K$. That is, if $R_{i,1:d}$ is our sample of rewards for prompt $x_i$ and $\hat{D}_i$ is our estimate of the reward distribution $r \circ \pi(x_i)$ constructed from $R_{i,1:d}$, then we would like to compute for each $j \in [(B-d)K]$, the scalar

$$V_{i,j} := \mathbb{E}_{Z_1,\ldots,Z_j \sim \hat{D}_i} \left[ \max\{R_{i,1}, \ldots, R_{i,d}, Z_1, \ldots, Z_j\} \right]. \tag{2}$$

In the last step, we use the vectors $\{V_i\}_{i \in [K]}$ to compute the remaining allocation $A \in [(B-d)K]^K$ that maximizes $\sum_{i=1}^{K} V_{i,A_i}$ under the constraint that $\sum_{i=1}^{K} A_i \leq (B-d)K$. We do so by using the greedy procedure in Algorithm 1, which is optimal (Federgruen and Groenevelt, 1986) if the vectors $V_1, \ldots, V_K$ are "concave" and monotonically increasing (i.e $V_{i,j+1} - V_{i,j} \geq V_{i,j+2} - V_{i,j+1}$ and $V_{i,j+1} \geq V_{i,j}$ ). Fortunately, Proposition 3.1, proved in Appendix E, shows that this is the case.

**Proposition 3.1.** *Let $D$ be any distribution with finite first moment and $c \in \mathbb{R}$ be some constant. Then, the function $f(n) = \mathbb{E}_{X_{1:n} \sim D^n} \left[ \max\{c, X_{1:n}\} \right]$ is concave and monotonically increasing.*

---

**Algorithm 1** Greedy Allocation

**Input:** Budget $T \in \mathbb{N}$, monotonically increasing, "concave" reward vectors $\{V_i\}_{i \in [K]}$
1 **Initialize**: $a = [0]^K$.
2 **for** $t = 1, \ldots, T$ **do**
3 $\quad$ Let $i_t \in \arg\max_{i \in [K]} (V_{i,a_i+1} - V_{i,a_i})$ and set $a_{i_t} = a_{i_t} + 1$
4 **end**
5 Return $a$.

---

In practice, we cannot compute Equation 2 exactly. Instead, we compute an estimate $\hat{V}_{i,j}$ of $V_{i,j}$ via Monte Carlo sampling from $\hat{D}_i$, which can be done very efficiently based on our choice of the reward distribution estimator in Section 3.1. While the greedy procedure may not be optimal when run on the estimated vectors, it still serves as an efficient heuristic. Moreover, Monte Carlo estimation of $V_{i,j}$ does not exhaust our total budget $BK$ as we no longer need to query the base LM.

---

**Algorithm 2** Two-stage Adaptive Best-of-$N$ (AdaBoN) Allocation Policy

**Input:** Per-prompt budget $B$, base LM $\pi$, reward function $r$, prompts $x_{1:K}$, per-prompt exploration budget $d$, estimation procedure $f$
1 For each $i \in [K]$, use $d$ inference calls to $\pi$ to obtain initial rewards $R_{i,1}, \ldots, R_{i,d}$.
2 For each $i \in [K]$, pass $R_{i,1}, \ldots, R_{i,d}$ to $f$ and obtain an estimate $\hat{D}_i$ of $r \circ \pi(x_i)$.
3 For each $i \in [K]$ and $j \in [(B-d)K]$, use Monte Carlo sampling to construct an estimate $\hat{V}_{i,j}$ of

$$V_{i,j} = \mathbb{E}_{Z_{i,1},\ldots,Z_{i,j} \sim \hat{D}_i} \left[ \max\{R_{i,1}, \ldots, R_{i,d}, Z_{i,1}, \ldots, Z_{i,j}\} \right].$$

4 Get remaining allocation $A$ by running Algorithm 1 with budget $(B-d)K$ and vectors $\{\hat{V}_i\}_{i \in [K]}$.

---

*A crucial property of AdaBoN is that it minimizes latency since calls to the base LM can be easily parallelized.* Indeed, only two calls to the base LM need to be made – the first call in the exploration

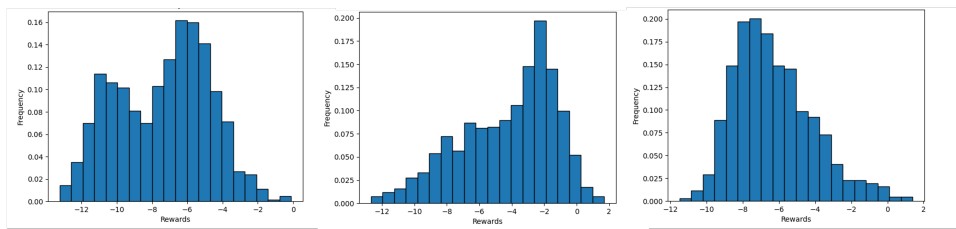

Figure 1: Reward distribution for three different prompts from the **AlpacaEval dataset** when responses are generated from Meta-Llama-3-8B and evaluated using FsfairX-LLaMA3-RM-v0.1. We provide reward distributions for the datasets in Appendix F.

.

stage and the second call once the remaining allocation has been determined. This is in contrast to existing work which design more adaptive policies (Manvi et al., 2024). What remains now is how to efficiently obtain an estimate of the reward distributions in Line 2 of Algorithm 2.

### 3.1 REWARD DISTRIBUTION ESTIMATION

To help guide our selection of estimation procedure in Algorithm 2, we plotted the histogram of samples from the reward distributions for several pairs of LMs, RMs, and prompts across the AlpacaEval, HH-RLHF, and PKU-SafeRLHF datasets. In Figure 1, we provide a few reward distribution when the LM is Meta-Llama-3-8B and the RM is FsfairX-LLaMA3-RM-v0.1 for the AlpacaEval dataset. Example reward distributions for the other datasets can be found in Appendix F. Across all LM-RM pairs we consider (see Section 4), we find that reward distributions are mostly smooth, have a few modes, and can be skewed. For such distributions, perhaps the simplest distribution estimation procedure is *kernel density estimation* (KDE) using a Gaussian kernel (Węglarczyk, 2018). In particular, given a sample of rewards $R_1^i, \ldots, R_d^i$ and a bandwidth parameter $h$, the Gaussian kernel density estimate returns the density function $\hat{f}_h(x) := \frac{1}{dh} \sum_{j=1}^d \phi\left(\frac{x - R_j^i}{h}\right)$, where $\phi$ is the density function of a standard normal random variable. To pick the bandwidth $h$, we use Scott's rule (Scott, 1979), a standard automatic bandwidth selection rule which sets $h = \hat{\sigma} d^{\frac{1}{5}}$, where $\hat{\sigma}$ is the sample standard deviation. To generate a sample according to a random variable with density $\hat{f}_h$, one first samples a reward $R \sim \{R_1^i, \ldots, R_d^i\}$ uniformly at random and then adds Gaussian noise with mean 0 and standard deviation $h$. Accordingly, $V_i$ can be estimated efficiently via Monte Carlo sampling.

Despite its simplicity, in Section 4 we show that Gaussian KDE using Scott's rule is remarkably *robust* – it is sufficient to consistently outperform our benchmarks across *all* LM-RM pairs. To compare, we also tried fitting Gaussian and Skew-Normal distributions using Maximum Likelihood Estimation (MLE). We present these results in Table 16 in Appendix K.3 and find that they performed worse than Gaussian KDE across most LM-RM pairs and datasets.

## 4 EXPERIMENTS

### 4.1 EXPERIMENTAL SETUP

**Datasets.** We consider three datasets, AlpacaEval, HH-RLHF and PKU-SafeRLHF, and achieve similar performance across all of them. For space reasons, we only present results for the AlpacaEval dataset in the main text. Results for the two other datasets, HH-RLHF and PKU-SafeRLHF, are in Appendix H. For each dataset, we construct $n = 50$ batches of size $K$ by sampling prompts uniformly at random without replacement from the total set of prompts. This ensures that all batches have distinct prompts. For each batch size $K$ we consider, we do this process once. The same collections of 50 batches is then used across all experiments for that batch size and dataset.

**Language and Reward Models.** We consider a range of LMs and RMs, all around 8B parameters. For LMs, we use Mistral-7B-v0.3, Gemma-7B, Qwen2.5-7B-Instruct, and Meta-Llama-3-8B. As for RMs, we focus on *real-valued* RMs. In particular, we use RM-Mistral-7B, FsfairX-LLaMA3-RM-v0.1, and ArmoRM-Llama3-8B-v0.1, all of which were also used by Sun et al. (2024).

### 4.2 EVALUATION METRICS AND BENCHMARKS

Although our allocation strategy is designed to maximize the cumulative sum of max rewards, the main objective we use for evaluation is the *Batch Win Rate (BWR)*. Formally, for a batch of prompts $x_1, .., x_K$, LM $\pi$, RM $r$, per-prompt inference budget $B \geq 1$, and an allocation policy $\mathcal{A}$, the batch win rate of $\mathcal{A}$ against the uniform allocation $a = [B, \ldots, B]$, is defined as

$$
\text{BWR}_{\mathcal{A}}(x_{1:K}, B) := \mathop{\mathbb{P}}_{\substack{R_{i,j} \sim r \circ \pi(x_i) \\ A \sim \mathcal{A}(\{R_{i,j}\}, B)}} \left[ \sum_{i=1}^{K} \max_{j=1,\ldots,A_i} R_{i,j} > \sum_{i=1}^{K} \max_{j=1,\ldots,B} R_{i,j} \right] +
$$

$$
\frac{1}{2} \cdot \mathop{\mathbb{P}}_{\substack{R_{i,j} \sim r \circ \pi(x_i) \\ A \sim \mathcal{A}(\{R_{i,j}\}, B)}} \left[ \sum_{i=1}^{K} \max_{j=1,\ldots,A_i} R_{i,j} = \sum_{i=1}^{K} \max_{j=1,\ldots,B} R_{i,j} \right]. \quad (3)
$$

This metric measures the probability, over both the random draws from the distributions $r \circ \pi(x_i)$ and $\mathcal{A}$, that our allocation beats the uniform allocation with the same inference budget. We weight the probability of a tie by $1/2$ to ensure that the BWR of the uniform allocation against itself is $0.50$. Hence, obtaining BWRs$> 0.50$ indicates *outperforming* the uniform allocation.

Our choice of the win rate over the expected cumulative max reward is because the scalar outputs of RMs are often only meaningful comparatively. That is, for a prompt $x$ and two responses $y_1, y_2 \in \mathcal{Y}$, the precise values of $r(x, y_1)$ and $r(x, y_2)$ are often meaningless, as they can be logits of a language model (Son et al., 2024; Ouyang et al., 2022; Christiano et al., 2017). On the other hand, the comparisons are meaningful as the RM is usually trained using preference data under the Bradley-Terry model (Bradley and Terry, 1952). Hence, $r(x, y_1) > r(x, y_2)$ tells us that the response $y_1$ is preferred over response $y_2$. Our benchmark of the uniform allocation is natural since, without knowledge of the true reward distributions, it is the minimax optimal *non-adaptive* allocation.

To get a better sense of the performance of AdaBoN, we also evaluate AdaBoN against uniform allocation strategies with strictly *larger* inference budgets. For a batch $x_{1:K}$, per-prompt budget $B$, and number $N \in \mathbb{N}$,

$$
\text{BWTR}_{\mathcal{A}}(x_{1:K}, N, B) := \mathop{\mathbb{P}}_{\substack{R_{i,j} \sim r \circ \pi(x_i) \\ A \sim \mathcal{A}(\{R_{i,j}\}, B)}} \left[ \sum_{i=1}^{K} \max_{j=1,\ldots,A_i} R_{i,j} \geq \sum_{i=1}^{K} \max_{j=1,\ldots,N} R_{i,j} \right] \quad (4)
$$

is the *batch win-tie rate*, i.e. the probability that $\mathcal{A}$, with a total inference budget of $BK$, does *at least as good* as the uniform allocation that allocates $N$ queries to each prompt in the batch. The main difference between Equations 3 and 4 is weighing the probability of a tie by 1 instead of a $1/2$. This is justified given that we will be competing against *larger* inference budgets.

One can also interpret $\text{BWTR}_{\mathcal{A}}(x_{1:K}, N, B)$ as the probability that AdaBoN *survives* against the uniform allocation with a per-prompt budget $N \in \mathbb{N}$. From this perspective,

$$
\text{S}_{\mathcal{A}}(x_{1:K}, B) := \sum_{N=1}^{\infty} \text{BWTR}_{\mathcal{A}}(x_{1:K}, N, B) \quad (5)
$$

$$
= \mathop{\mathbb{E}}_{\substack{R_{i,j} \sim r \circ \pi(x_i) \\ A \sim \mathcal{A}(\{R_{i,j}\}, B)}} \left[ \arg\max_{N \in \mathbb{N}} \left\{ \sum_{i=1}^{K} \max_{j=1,\ldots,A_i} R_{i,j} \geq \sum_{i=1}^{K} \max_{j=1,\ldots,N} R_{i,j} \right\} \right]
$$

can be interpreted as the *Expected Survival Time (EST)* (Jenkins, 2005) for batch $x_{1:K}$. Larger ESTs indicate that AdaBoN with a per-prompt budget of $B$ is competitive against uniform allocations with larger budgets. Thus, larger ESTs imply *computational savings* – to obtain guarantees comparable to having an inference budget of size $\text{S}_{\mathcal{A}}(x_{1:K}, B) \cdot K$, one needs an inference budget of size $BK$.

Lastly, although Damani et al. (2024) study the same allocation problem, we do not compare with their approach for the following reasons. First, we were unable to find an existing implementation of their method or sufficient details about hyperparameter choices to implement their method faithfully. Second, the approach by Damani et al. (2024) is more computationally demanding. It requires training a separate MLP for each LM-RM pair and each value of $b \in [BK]$. For our experiments, we set $K = 5$, $B = 120$ and consider 12 LM-RM pairs and 3 datasets. This results in needing to train $216,000$ MLPs, which is computationally prohibitive.

Table 1: Median [Q1, Q3] BWRs for $K = 5$, $B = 120$, and $d = 0.75B$ on the **AlpacaEval dataset**.

| LM | RM | | |
|---|---|---|---|
| | Mistral | FsfairX | Armo |
| Mistral | 0.58 [0.57, 0.60] | 0.58 [0.55, 0.60] | 0.59 [0.55, 0.62] |
| Qwen | 0.60 [0.59, 0.63] | 0.62 [0.59, 0.65] | 0.54 [0.51, 0.56] |
| Gemma | 0.56 [0.51, 0.59] | 0.55 [0.54, 0.59] | 0.56 [0.53, 0.58] |
| Llama | 0.58 [0.54, 0.63] | 0.59 [0.55, 0.62] | 0.59 [0.56, 0.62] |

Table 2: (a) Median [Q1, Q3] EST for $K = 5$, $B = 120$, and $d = 0.75B$ and (b) Percent batches with BWR $> 0.50$ for $K = 5$, $B = 120$, and $d = 0.75B$, both for the **AlpacaEval dataset**.

(a)

| LM | RM | | |
|---|---|---|---|
| | Mistral | FsfairX | Armo |
| Mistral | 151 [148, 152] | 150 [148, 152] | 151 [150, 155] |
| Qwen | 152 [150, 154] | 151 [150, 154] | 153 [150, 156] |
| Gemma | 148 [146, 150] | 148 [147, 151] | 149 [147, 151] |
| Llama | 151 [148, 153] | 151 [148, 153] | 151 [149, 154] |

(b)

| LM | RM | | |
|---|---|---|---|
| | Mistral | FsfairX | Armo |
| Mistral | 94% | 92% | 96% |
| Qwen | 100% | 98% | 78% |
| Gemma | 76% | 92% | 86% |
| Llama | 92% | 96% | 92% |

## 4.3 MAIN RESULTS

In this section, we mostly present results for $K = 5$, $B = 120$, and $d = 0.75B$. We estimate the EST by capping the sum in Equation 5 to $2B$. In Appendix K, we perform ablations where we test the performance against various choices of $K$ and $B$, keeping $d = 0.75B$ fixed. For our reward distribution estimation procedure, we use Gaussian kernel density estimation and pick the bandwidth using Scott's rule, as described in Section 3.1. To estimate the $V_{i,j}$'s in Line 4 of Algorithm 2, we use Monte Carlo sampling with a sample size $m = 1024$. For the sake of replicability, we use the standard generation function from Hugging Face (Wolf et al., 2019), and thus use the default decoding strategy for all LMs. For a batch of prompts, we estimated its BWR (Equation 3) and EST (Equation 5) by taking the average of the empirical BWR and empirical survival times over 100 runs. Due to space constraints, we only present the results for the AlpacaEval dataset in the main text. We find similar results for the HH-RLHF and PKU-SafeRLHF datasets in Appendix H.

**AdaBoN consistently and often significantly outperforms the uniform allocation.** Table 2b shows that AdaBoN outperforms the uniform allocation for more than $75\%$ of the batches across all LM-RM pairs. The performance is most prominent for the Qwen-Mistral pair, *where AdaBoN achieves a BWR $> 0.50$ for all batches.* Moreover, Table 1 shows that for many LM-RM pairs, AdaBoN actually achieves BWRs $\gtrsim 0.60$ on $50\%$ of the batches. Figure 2a provides a more fine-grained view of the *distribution* of BWRs for the Qwen-Instruct LM. *We find that on some batches AdaBoN achieves BWRs as high as $0.70$.* Despite its strong performance on the Qwen-Mistral and Qwen-FsfairX pairs, Figure 2a also shows that the performance of AdaBoN drops for Qwen-Armo. We explain this phenomena in Appendix G.1 by showing that the vast majority of reward distributions for this LM-RM pair are *left-skewed*. Results for the other LM-RM pairs are in Appendix G.

**AdaBoN is competitive against larger inference budgets.** Table 2a gives the median [Q1, Q3] ESTs for AdaBoN across the 50 batches. AdaBoN obtains comparable performance against uniform allocations with $20\%$ larger inference budget. Because batches vary in difficulty, we consider the distribution of ESTs across the 50 batches. Figure 2b gives a box-plot of the ESTs for the Qwen LM. We observe ESTs $\geq 160$, meaning that for these batches, *AdaBoN is competitive with unif. allocations at $33\%$ larger inference budget.* ESTs for other LM-RM pairs are in Appendix G.2.

**AdaBoN performs better as batch size increases.** In Appendix K.2, we fix $B = 120$ and $d = 0.75B$, and vary $K \in \{3, 5, 10, 15, 20\}$. Figure 3 shows that for all LM-RM pairs, the average BWR increases as $K$ increases from 3 to 20. *For some LM-RM pairs, like Qwen-Mistral, the average BWR*

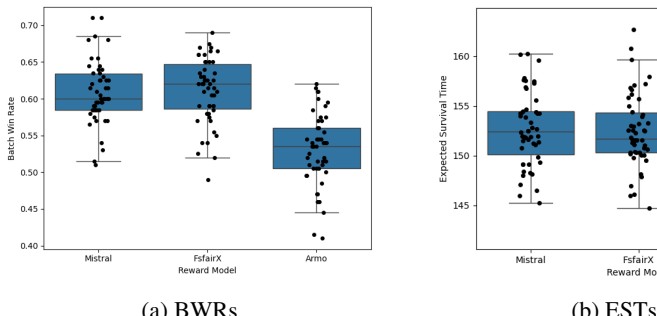

|          |          |
|:--------:|:--------:|
| (a) BWRs | (b) ESTs |

Figure 2: Box plots of BWRs and ESTs across the 50 batches for the Qwen-Instruct LM when $K = 5$, $B = 120$, and the exploration budget $d = 0.75B$ on the **AlpacaEval dataset**.

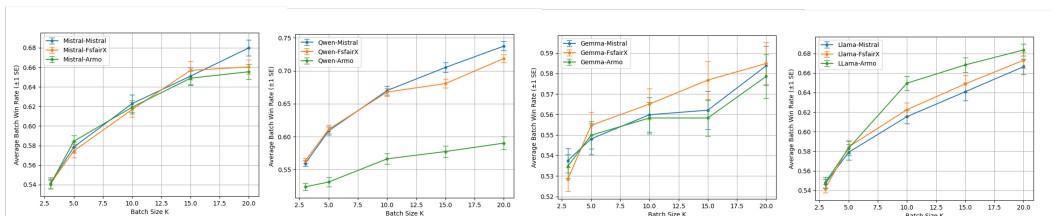

Figure 3: Avg. BWR ($\pm 1$ SE) as a function of $K \in \{3, 5, 10, 15, 20\}$ when $B = 120$ and $d = 0.75B$ on the **AlpacaEval dataset**.

*increases by as much as* $0.15$. Finally, Table 14 in Appendix K.2 shows that across most pairs of LM-RM, the percent of batches with BWR $> 0.50$ increases with $K$. The results for the Mistral LM are striking – *when $K = 20$, AdaBoN achieves a BWR $> 0.50$ for $100\%$ of batches for every RM.*

**AdaBoN maintains performance across inference budgets.** In Appendix K.1, we fix $K = 5$, $d = 0.75B$, and vary $B \in \{80, 100, 120, 140, 160\}$. Figure 9 shows that for all LM-RM pairs, the average BWR of AdaBoN generally increases with $B$, albeit modestly. The modest gain in performance is expected as the uniform allocation gets powerful with larger $B$ (see Appendix D).

**AdaBoN requires minimal hyperparameter tuning.** A notable feature of AdaBoN is that it requires minimal hyperparameter tuning. By using Gaussian kernel density estimation with an automatic bandwidth selector, *there is only one hyperparameter – the exploration budget $d$.* We find that simply fixing $d = 0.75B$ is a good initial guess. Table 3 in Appendix G.1 presents the median BWR across the 50 batches after tuning the exploration budget $d \in \{0.60B, 0.7B, 0.75B, 0.80B\}$ to maximize the median BWR. We find that that setting the exploration budget to $d = 0.75B$ incurs a minimal drop in median BWR compared to the optimal choice of exploration budget.

## 5 DISCUSSION AND LIMITATIONS

This work revisits Best-of-$N$ sampling and demonstrates that significant efficiency gains can be achieved through prompt-adaptive allocation of the sampling budget. There are several limitations to our approach. First, our method assumes that Gaussian kernel density estimation can sufficiently estimate the reward distributions. This may not hold for discrete RMs. An interesting future direction is to better understand the choice of reward estimation procedure on AdaBoN. Second, while our two-stage procedure is simple and effective, it does not dynamically refine its estimates during allocation. A more sophisticated bandit-based method could potentially improve performance guarantees, at the cost of increased latency. Finally, our method assumes access to a batch of prompts, making it less suitable for purely single-prompt settings. As such, it is an interesting future direction to study our setup in the *online* setting, where prompts arrive sequentially.

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

## A  DISCLOSURE OF LLM USAGE

LLMs were only used to aid and polish the writing in the Introduction, Related Works, and Discussion sections.

## B  OTHER RELATED WORK

Beyond input-adaptive inference allocation, prior work has also explored adaptivity at different granularities.

Wang et al. (2024b) propose Difficulty-Adaptive Self-Consistency (DSC), a cost-efficient decoding method for reasoning tasks that allocates sample budgets based on estimated query difficulty using both prior and posterior signals. Unlike DSC, which focuses on difficulty-adaptive sampling for reasoning tasks like arithmetic and commonsense QA, our work tackles inference-time alignment in open-ended generation by allocating queries based on learned reward distributions rather than problem difficulty.

Manvi et al. (2024) introduce capability-aware and mid-generation self-evaluations, allowing LMs to decide,during or after generation,whether further sampling would yield better outputs, thereby reducing compute without external reward models. Unlike this approach, which relies on self-evaluation to adaptively terminate or continue sampling per prompt, our method uses a learned model of reward distributions to allocate a fixed budget across prompts, focusing on batch-level optimization in open-ended generation.

Finally, Zhang et al. (2024a) propose OSCA, a method for optimizing how inference-time compute is distributed across a set of sampling configurations (e.g., temperature, model, prompt), aiming to improve pass rates under tight compute budgets across coding and reasoning tasks. In contrast, our work focuses on allocating a fixed compute budget across prompts, not configurations, based on learned reward distributions, enabling prompt-adaptive inference in open-ended generation tasks rather than pass@$k$ accuracy in structured problem-solving.

These methods target different axes of adaptivity but do not address how to allocate a fixed budget across multiple prompts. In contrast, we focus on the cross-prompt budget allocation problem, aiming to maximize the sum of per-prompt maxima while retaining the low-latency, parallelizable structure of Best-of-$N$ sampling.

## C  DATASET AND MODEL ASSET DETAILS

**Datasets.**  We consider three datasets:

- **AlpacaEval (v2.0)**: `https://github.com/tatsu-lab/alpaca_eval`, CC-BY-NC-4.0 license.
- **HH-RLHF**: `https://huggingface.co/datasets/Anthropic/hh-rlhf`, MIT License.
- **PKU-SafeRLHF**: `https://huggingface.co/datasets/PKU-Alignment/PKU-SafeRLHF`,CC-BY-NC-4.0 license.

We did not perform any additional data scraping. For each dataset separately, we construct 50 batches of prompts per batch size setting using uniform random sampling without replacement.

**Language Models (LMs).**  We use the following publicly available language models:

- **Mistral-7B-v0.3**: `https://huggingface.co/mistralai/Mistral-7B-v0.3`, Apache 2.0 License.
- **Gemma-7B**: `https://huggingface.co/google/gemma-7b`, Gemma License (non-commercial).
- **Qwen2.5-7B-Instruct**: `https://huggingface.co/Qwen/Qwen2.5-7B-Instruct`, Apache 2.0 License.

- **Meta-Llama-3-8B**: `https://huggingface.co/meta-llama/Meta-Llama-3-8B`, Llama 3 Community License.

**Reward Models (RMs).** We use externally provided real-valued reward models:

- **RM-Mistral-7B**: `https://huggingface.co/weqweasdas/RM-Mistral-7B`.
- **FsfairX-LLaMA3-RM-v0.1**: `https://huggingface.co/sfairXC/FsfairX-LLaMA3-RM-v0.1`, CC-BY-NC-4.0 License
- **ArmoRM-Llama3-8B-v0.1**: `https://huggingface.co/RLHFlow/ArmoRM-Llama3-8B-v0.1`, Llama 3 Community License

For all models and datasets, we follow their licensing terms and acknowledge the original sources.

# D  IMPACT OF INCREASING PER-PROMPT INFERENCE BUDGET $B$ ON BWR

In this section, we corroborate our claim in Section 4.3 that the uniform allocation gets more powerful as $B$ increases. To see this, consider the same example considered in Section 2.3. Brute force computation showed that when $B = 25$ and $d = 0.20B = 5$, the expected reward of the uniform allocation was 1.72 while the expected reward of the simple two-stage allocation procedure was 1.87.

Now, if one considers $B = 50$ and keeps $d = 0.20B$, then brute force computation shows that the expected reward of the uniform allocation is 1.92 while the expected reward of the simple two-stage allocation procedure is only 1.98. Notice that the *gap* between the expected reward of the simple two-stage allocation procedure and the uniform allocation has decreased as $B$ increased. This highlights the fact that the uniform allocation gets relatively more powerful as $B$ increases.

# E  PROOF OF PROPOSITION 3.1

*Proof.* Let $D$ be any distribution with finite first moment and $c \in \mathbb{R}$. Consider the function $f(n) = \mathbb{E}_{X_{1:n} \sim D^n} [\max\{c, X_{1:n}\}]$. We first show that $f$ is monotonically non-decreasing. It suffices to show that $f(n) \geq f(n-1)$ for all $n \geq 2$. Fix some $n \geq 2$ and define the random variable $M_n = \max\{c, X_{1:n}\}$, where $X_{1:n} \sim D^n$. Then, observe that $M_n \geq M_{n-1}$ pointwise for every realization of random variables $X_{1:n} \sim D^n$. Taking expectations of both sides, gives that $f(n) \geq f(n-1)$, completing this part of the proof.

We now prove that $f$ is "concave". For $n \geq 2$, define $\Delta_n := f(n) - f(n-1)$. It suffices to show that $\Delta_{n+1} \leq \Delta_n$ for all $n \geq 2$. Fix some $n \geq 2$. Observe that we can write

$$\Delta_n = \mathbb{E}_{X_{1:n} \sim D^n} [M_n - M_{n-1}] = \mathbb{E}_{X_{1:n} \sim D^n} [(X_n - M_{n-1})_+]$$

where $(x)_+ = \max(x, 0)$. Likewise, we can write

$$\Delta_{n+1} = \mathbb{E}_{X_{1:n+1} \sim D^{n+1}} [(X_{n+1} - M_n)_+].$$

Hence, we need to show that

$$\mathbb{E}_{X_{1:n+1} \sim D^{n+1}} [(X_{n+1} - M_n)_+] \leq \mathbb{E}_{X_{1:n} \sim D^n} [(X_n - M_{n-1})_+].$$

Since $M_n = \max\{X_n, M_{n-1}\}$ and $X_n \sim D$, we have that $M_n \geq M_{n-1}$ pointwise for every realization of random variables $X_{1:n} \sim D^n$. Thus, pointwise for any $x \in \mathbb{R}$ and realization of random variables $X_{1:n} \sim D^n$, we have that

$$(x - M_n)_+ \leq (x - M_{n-1})_+.$$

Taking expectations of both sides, we have that

$$\mathbb{E}_{X \sim D, X_{1:n} \sim D^n} [(X - M_n)_+] \leq \mathbb{E}_{X \sim D, X_{1:n-1} \sim D^{n-1}} [(X - M_{n-1})_+].$$

Finally, noting that

$$\mathbb{E}_{X \sim D, X_{1:n} \sim D^n} [(X - M_n)_+] = \mathbb{E}_{X_{1:n+1} \sim D^{n+1}} [(X_{n+1} - M_n)_+],$$

and

$$\mathbb{E}_{X \sim D, X_{1:n-1} \sim D^{n-1}} [X - M_{n-1})_+] = \mathbb{E}_{X_{1:n} \sim D^n} [(X_n - M_{n-1})_+]$$

completes the proof. ∎

## F    REWARD DISTRIBUTIONS FOR HH-RLHF AND PKU-SAFERLHF DATASETS

In this section, we provide some plots of reward distributions for Meta-Llama-3-8B and and FsfairX-LLaMA3-RM-v0.1 for prompts from the HH-RLHF and PHU-SafeRLHF datasets respectively. Like AlpacaEval, we observe that the reward distributions are general smooth and amenable to Gaussian KDE.

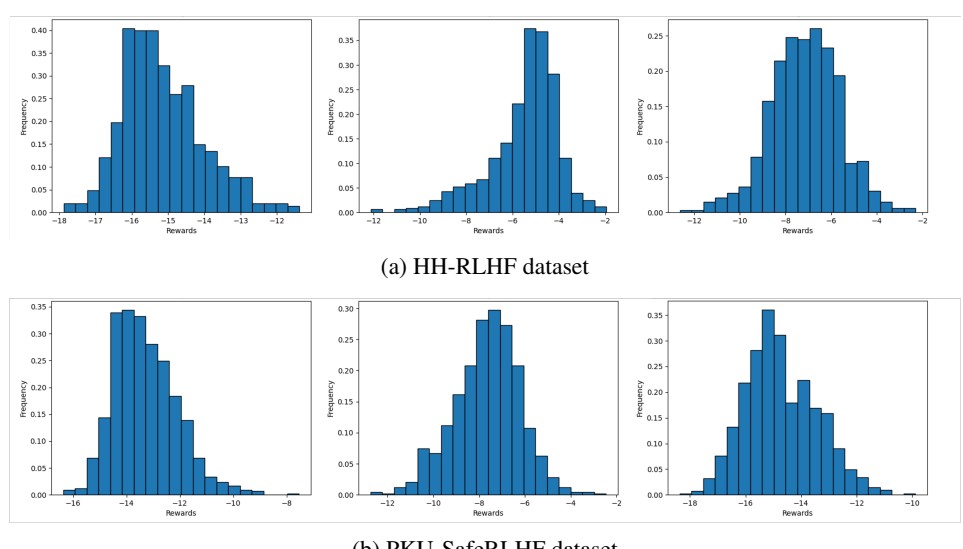

(a) HH-RLHF dataset

(b) PKU-SafeRLHF dataset

Figure 4: Reward distributions for three different prompts when responses are generated from Meta-Llama-3-8B and evaluated using FsfairX-LLaMA3-RM-v0.1.

## G    MISSING TABLES AND FIGURES FOR ALPACAEVAL DATASET

In this section, we provide the missing tables and figures from the main text for the AlpacaEval Dataset.

### G.1    BATCH WIN RATES

In this section, we provide box-plots of the BWRs for the remaining LM-RM pairs when $K = 5$, $B = 120$, and $d = 0.75B$ for the AlpacaEval dataset. We find that across all LM-RM pairs, AdaBoN

achieves a BWR > 0.50 for the vast majority of batches. Moreover, AdaBoN consistently achieves BWRs larger than 0.60 for ≈ 25% of batches for all LM-RM pairs.

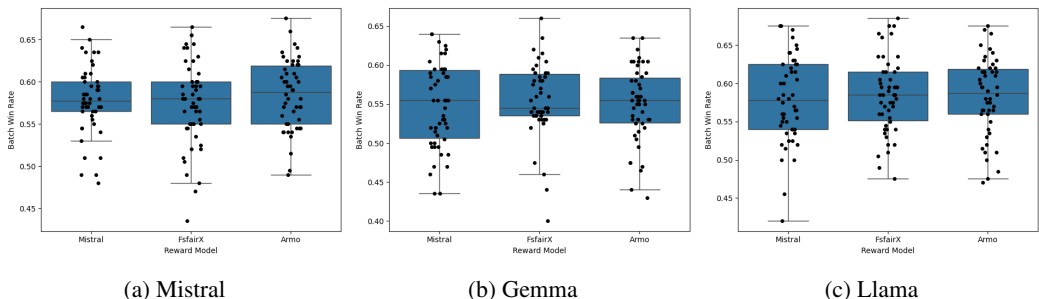

(a) Mistral          (b) Gemma          (c) Llama

Figure 5: Box plots of BWRs for batch size $K = 5$, inference budget $B = 120$ and the exploration budget $d = 0.75B$ on the **AlpacaEval dataset**.

Table 3 provides the BWR for $K = 5$ and $B = 120$, when the exploration budget $d$ is optimized between $\{0.60B, 0.70B, 0.75B, 0.80B\}$. Here, we find that setting the exploration budget to $d = 0.75B$ is a good guess as the optimized median BWR is not too much higher across all LM-RM pairs.

Table 3: Median BWR for batch size $K = 5$, inference budget $B = 120$, and exploration budget $d$ optimized between $\{0.60B, 0.70B, 0.75B, 0.80B\}$ to maximize the median dataset BWR on the **AlpacaEval dataset**.

| **LM** | **RM** | | |
|---|---|---|---|
| | Mistral | FsfairX | Armo |
| Mistral | 0.58 | 0.58 | 0.59 |
| Qwen | 0.60 | 0.63 | 0.54 |
| Gemma | 0.56 | 0.56 | 0.56 |
| Llama | 0.59 | 0.59 | 0.59 |

Table 3 shows that the Qwen LM exhibits a significant drop in median BWR between the Mistral/Fs-fairX RMs and the Armo RM. To investigate this, we plotted the reward distribution for the Qwen-Armo LM-RM pair across several prompts. Compared to the Qwen-Mistral and Qwen-FsfariX LM-RM pairs, we find that the reward distributions for the Qwen-Armo LM-RM pair are *significantly left skewed*. This makes adaptivity less useful as the uniform allocation for batches with left-skewed reward distributions is close to optimal. To capture this, Figure 6 plots a histogram of the skewness (i.e. Pearson's moment coefficient of skewness) of the reward distributions across all prompts for the Qwen LM.

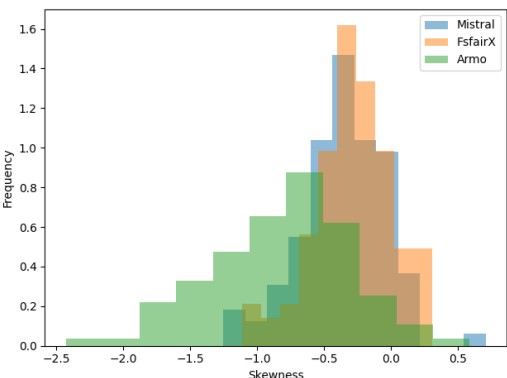

Figure 6: Histograms of skewness (i.e. Pearson's moment coefficient of skewness) of reward distributions for the Qwen LM across all prompts from the **AlpacaEval dataset**. We observe that the reward distributions for the Qwen-Armo LM-RM pair is significantly more left-skewed than Qwen-Mistral or Qwen-FsfairX. In fact, we find that the reward distributions for the vast majority of prompts are left-skewed for the Qwen-Armo pair.

From here, we observe that indeed the reward distributions for the Qwen-Armo LM-RM pair are significantly more left-skewed than the reward distributions of the Qwen-Mistral and Qwen-FsfairX LM-RM pair. We provide the histogram of the skewness for the remaining LMs in Figure 7.

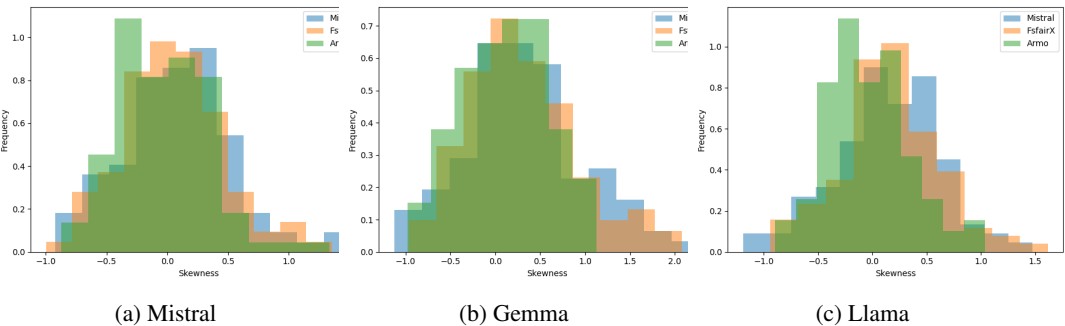

(a) Mistral          (b) Gemma          (c) Llama

Figure 7: Histograms of skewness (i.e. Pearson's moment coefficient of skewness) of reward distributions for the Mistral, Gemma, and Llama LM on prompts from the **AlpacaEval dataset**. Unlike the Qwen LM, we observe that the skewness of the reward distributions for the other LMs do not deviate significantly between RMs.

Compared to the Qwen LM, for the remaining LMs, we find that the histograms of skewness across the reward distributions do not vary significantly between different RMs. This corroborates our results in Table 3, which shows that the optimal median BWR is roughly the same across all RMs for the Mistral, Gemma, and Llama LM.

## G.2 EXPECTED SURVIVAL TIMES

Figure 8 provides the box-plots of ESTs for the remaining pairs of LM and RMs for $K = 5$, $B = 120$, and $d = 0.75B$.

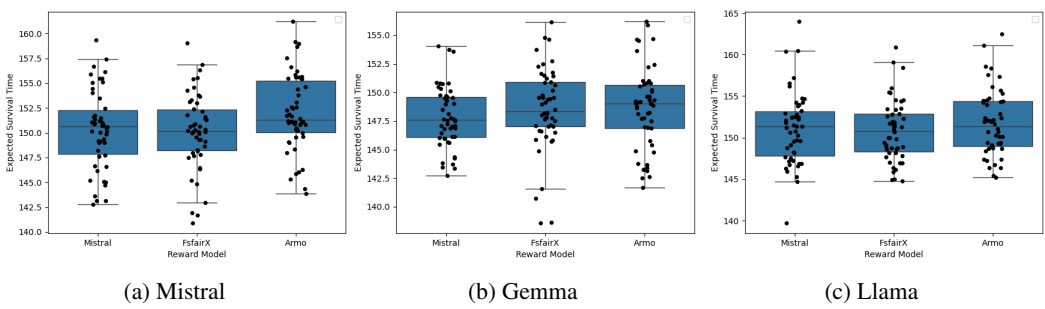

(a) Mistral        (b) Gemma        (c) Llama

Figure 8: Box plot of ESTs for batch size $K = 5$, budget $B = 120$ and exploration budget $d = 0.75B$ on the **AlpacaEval dataset**.

## H  RESULTS FOR THE HH-RLHF AND PKU-SAFERLHF DATASETS

In this section, we present our experimental results for the HH-RLHF and PKU-SafeRLHF datasets, where we produce tables similar to Tables 1, 2a, and 2b in the main paper. Overall, we find that the results resemble those for the AlpacaEval dataset for both BWRs and ESTs. Like the AlpacaEval dataset, we find that the performance for the Qwen-Armo LM-RM pair is significantly lower than the other LM-RM pairs for both datasets. Again, we found that for this LM-RM pair, the majority of its reward distributions are left-skewed.

Table 4: Median [Q1, Q3] BWRs for $K = 5$, $B = 120$, and $d = 0.75B$ on the **HH-RLHF dataset**.

| LM | RM | | |
|---|---|---|---|
| | Mistral | FsfairX | Armo |
| Mistral | $0.58\,[0.55, 0.61]$ | $0.60\,[0.57, 0.62]$ | $0.55\,[0.53, 0.59]$ |
| Qwen | $0.55\,[0.52, 0.58]$ | $0.54\,[0.51, 0.57]$ | $0.53\,[0.50, 0.55]]$ |
| Gemma | $0.54\,[0.49, 0.57]$ | $0.53\,[0.47, 0.56]$ | $0.55\,[0.51, 0.57]$ |
| Llama | $0.59\,[0.56, 0.61]$ | $0.57\,[0.53, 0.59]$ | $0.57\,[0.55, 0.60]$ |

Table 5: (a) Median [Q1, Q3] EST for $K = 5$, $B = 120$, and $d = 0.75B$. (b) Percent batches with BWR $> 0.50$ for $K = 5$, $B = 120$, and $d = 0.75B$ on the **HH-RLHF dataset**.

(a)

| LM | RM | | |
|---|---|---|---|
| | Mistral | FsfairX | Armo |
| Mistral | $151\,[146, 163]$ | $151\,[146, 169]$ | $151\,[146, 167]$ |
| Qwen | $148\,[141, 182]$ | $150\,[143, 222]$ | $154\,[147, 221]$ |
| Gemma | $149\,[143, 162]$ | $149\,[143, 157]$ | $150\,[144, 227]$ |
| Llama | $153\,[148, 182]$ | $150\,[144, 162]$ | $154\,[146, 183]$ |

(b)

| LM | RM | | |
|---|---|---|---|
| | Mistral | FsfairX | Armo |
| Mistral | 94% | 94% | 92% |
| Qwen | 80% | 76% | 72% |
| Gemma | 70% | 54% | 76% |
| Llama | 98% | 82% | 92% |

Table 6: Median [Q1, Q3] BWRs for $K = 5$, $B = 120$, and $d = 0.75B$ on the **PKU-SafeRLHF dataset**.

| LM | RM | | |
|---|---|---|---|
| | Mistral | FsfairX | Armo |
| Mistral | $0.57\,[0.55, 0.60]$ | $0.57\,[0.53, 0.60]$ | $0.57\,[0.55, 0.61]$ |
| Qwen | $0.54\,[0.53, 0.57]$ | $0.56\,[0.52, 0.60]$ | $0.49\,[0.46, 0.51]$ |
| Gemma | $0.53\,[0.50, 0.58]$ | $0.58\,[0.55, 0.60]$ | $0.57\,[0.53, 0.60]$ |
| Llama | $0.56\,[0.52, 0.59]$ | $0.59\,[0.54, 0.62]$ | $0.61\,[0.58, 0.62]$ |

Table 7: (a) Median [Q1, Q3] EST for $K = 5$, $B = 120$, and $d = 0.75B$ and (b) Percent batches with BWR $> 0.50$ for $K = 5$, $B = 120$, and $d = 0.75B$, both for the **PKU-SafeRLHF dataset**.

(a)

| LM | RM | | |
|---|---|---|---|
| | Mistral | FsfairX | Armo |
| Mistral | $151\,[145, 179]$ | $152\,[146, 190]$ | $151\,[146, 228]$ |
| Qwen | $153\,[147, 211]$ | $151\,[144, 197]$ | $152\,[146, 234]$ |
| Gemma | $148\,[145, 180]$ | $151\,[146, 210]$ | $150\,[143, 182]$ |
| Llama | $151\,[144, 163]$ | $152\,[145, 182]$ | $154\,[148, 197]$ |

(b)

| LM | RM | | |
|---|---|---|---|
| | Mistral | FsfairX | Armo |
| Mistral | $96\%$ | $92\%$ | $88\%$ |
| Qwen | $88\%$ | $80\%$ | $38\%$ |
| Gemma | $74\%$ | $96\%$ | $90\%$ |
| Llama | $78\%$ | $94\%$ | $98\%$ |

# I COMPARISON TO VARIANCE-BASED ADAPTIVE BASELINE

In this section, we benchmark the performance of AdaBoN against a simple variance-based adaptive allocation policy we call VarBoN. VarBoN operates as follows. Similar to AdaBoN, VarBoN fixes an exploration budget $d = 0.75B$, and samples $d$ responses and rewards for each prompt in the batch. Let $\{R_{i,j}\}_{i \in [K], j \in [d]}$ denote the corresponding set of rewards, where $R_{i,j}$ denotes the the $j$'th reward for the $i$'th prompt in the batch. Then, for each prompt $i \in [K]$, VarBoN computes the empirical standard deviation of $R_{i,1:d}$ which denote by $\hat{\sigma}_i$. Finally, VarBoN constructs a distribution $\pi$ over $[K]$ such that $\pi_i = \frac{\hat{\sigma}_i}{\sum_i \hat{\sigma}_i}$ and allocates a $\pi_i$ fraction of the remaining budget $(B - d)K$ to prompt $i$. In other words, the remaining inference budget is allocated proportionally to the empirical standard deviation of rewards obtained during the exploration phase. Tables 8 and 9 compare the BWR of AdaBoN against VarBoN and VarBoN against the uniform allocation respectively, for the AlpacaEval dataset.

Table 8: Median [Q1, Q3] BWR of AdaBoN vs VarBoN for $K = 5$, $B = 120$, and $d = 0.75B$ on the **AlpacaEval dataset**.

| LM | RM | | |
|---|---|---|---|
| | Mistral | FsfairX | Armo |
| Mistral | $0.58\,[0.56, 0.61]$ | $0.60\,[0.58, 0.63]$ | $0.59\,[0.56, 0.61]$ |
| Qwen | $0.56\,[0.53, 0.59]$ | $0.55\,[0.52, 0.58]$ | $0.54\,[0.51, 0.56]$ |
| Gemma | $0.57\,[0.54, 0.60]$ | $0.57\,[0.54, 0.61]$ | $0.55\,[0.51, 0.59]$ |
| Llama | $0.59\,[0.56, 0.62]$ | $0.60\,[0.56, 0.62]$ | $0.60\,[0.57, 0.62]$ |

Table 9: Median [Q1, Q3] BWR of VarBoN vs Uniform Allocation for $K = 5$, $B = 120$, and $d = 0.75B$ on the **AlpacaEval dataset**.

| LM | RM | | |
|---|---|---|---|
| | Mistral | FsfairX | Armo |
| Mistral | 0.49 [0.48, 0.50] | 0.49 [0.48, 0.50] | 0.49 [0.48, 0.50] |
| Qwen | 0.49 [0.46, 0.50] | 0.48 [0.47, 0.50] | 0.50 [0.48, 0.51] |
| Gemma | 0.48 [0.46, 0.49] | 0.50 [0.49, 0.51] | 0.48 [0.47, 0.51] |
| Llama | 0.50 [0.47, 0.51] | 0.49 [0.48, 0.50] | 0.48 [0.46, 0.49] |

We find that VarBoN performs comparably to the uniform allocation, but worse than AdaBoN across all LLM-RM pairs.

## J  PER-PROMPT WIN RATES

In this work, our main evaluation metric is the BWR, which compares the sum of the rewards across the batch of prompts. This is natural given that our optimization objective, stated in Equation 1, is the cumulative sum of rewards across the batch of prompts. However, in practice, the *per-prompt* win rate is also important to ensure that our performance is not too bad for any particular prompt. Given a batch of prompts $x_{1:K}$ and a per-prompt inference budget $B$, we define the average per-prompt win rate as

$$\mathrm{WTR}_{\mathcal{A}}(x_{1:K}, B) := \frac{1}{K} \sum_{i=1}^{K} \mathop{\mathbb{P}}_{\substack{R_{i,j} \sim r \circ \pi(x_i) \\ A \sim \mathcal{A}(\{R_{i,j}\}, B)}} \left[ \max_{j=1,\dots,A_i} R_{i,j} \geq \max_{j=1,\dots,B} R_{i,j} \right].$$

In Table 10, we give the Median [Q1, Q3] WTR of AdaBoN across 50 batches of prompts from the AlpacaEval dataset.

Table 10: Median [Q1, Q3] WTR of AdaBoN for $K = 5$, $B = 120$, and $d = 0.75B$ on the **AlpacaEval dataset**.

| LM | RM | | |
|---|---|---|---|
| | Mistral | FsfairX | Armo |
| Mistral | 0.51 [0.51, 0.52] | 0.52 [0.51, 0.52] | 0.52 [0.51, 0.52] |
| Qwen | 0.51 [0.50, 0.52] | 0.51 [0.50, 0.52] | 0.51 [0.50, 0.51] |
| Gemma | 0.51 [0.51, 0.52] | 0.51 [0.51, 0.52] | 0.51 [0.50, 0.52] |
| Llama | 0.52 [0.51, 0.52] | 0.52 [0.51, 0.52] | 0.52 [0.51, 0.52] |

From here, we find that despite AdaBoN only optimizing for the cumulative sum of rewards (and hence the BWTR), it is still competitive with the uniform allocation on a per-prompt basis.

## K  ABLATIONS

In this section, we sweep over choices of $B$ and $K$. We keep our choice of exploration budget $d = 0.75B$ fixed throughout and focus only on the AlpacaEval dataset.

### K.1  VARYING BUDGET $B$

Keeping $K = 5$ fixed, we consider budgets $B \in \{80, 100, 120, 140, 160\}$. Since the result for $B = 120$ is presented in the main text, we only present the results for the other four choices of $B$. Table 11 summarizes the results and showcases that AdaBoN continues to outperform uniform allocations at larger and smaller budget.

Table 11: Percent of batches with BWR $> 0.50$ for budgets $B \in \{80, 100, 140, 160\}$ on the **AlpacaEval dataset**, fixing batch size $K = 5$, and exploration budget $d = 0.75B$.

| LM | RM | | |
|---|---|---|---|
| | Mistral | FsfairX | Armo |
| **(a)** $B = 80$ | | | |
| Mistral | 96% | 96% | 92% |
| Qwen | 90% | 98% | 66% |
| Gemma | 68% | 78% | 54% |
| Llama | 86% | 96% | 90% |
| **(b)** $B = 100$ | | | |
| Mistral | 98% | 100% | 98% |
| Qwen | 100% | 100% | 72% |
| Gemma | 80% | 84% | 74% |
| Llama | 96% | 94% | 100% |
| **(c)** $B = 140$ | | | |
| Mistral | 92% | 98% | 98% |
| Qwen | 100% | 100% | 60% |
| Gemma | 68% | 82% | 82% |
| Llama | 90% | 94% | 100% |
| **(d)** $B = 160$ | | | |
| Mistral | 100% | 100% | 94% |
| Qwen | 98% | 100% | 86% |
| Gemma | 80% | 92% | 78% |
| Llama | 90% | 98% | 100% |

Table 12: Median [Q1, Q3] BWRs for budgets $B \in \{80, 100, 140, 160\}$ on the **AlpacaEval dataset**, fixing batch size $K = 5$ and exploration budget $d = 0.75B$.

| LM | RM | | |
|---|---|---|---|
| | Mistral | FsfairX | Armo |
| **(a)** $B = 80$ | | | |
| Mistral | 0.58[0.54, 0.61] | 0.57[0.55, 0.60] | 0.56[0.53, 0.59] |
| Qwen | 0.57[0.54, 0.59] | 0.59[0.56, 0.61] | 0.52[0.48, 0.55] |
| Gemma | 0.53[0.49, 0.56] | 0.54[0.51, 0.58] | 0.51[0.49, 0.55] |
| Llama | 0.55[0.52, 0.59] | 0.58[0.55, 0.60] | 0.57[0.52, 0.60] |
| **(b)** $B = 100$ | | | |
| Mistral | 0.59[0.57, 0.61] | 0.59[0.56, 0.60] | 0.59[0.56, 0.61] |
| Qwen | 0.57[0.55, 0.59] | 0.60[0.56, 0.62] | 0.53[0.50, 0.55] |
| Gemma | 0.54[0.51, 0.58] | 0.56[0.52, 0.59] | 0.55[0.50, 0.58] |
| Llama | 0.56[0.53, 0.59] | 0.57[0.55, 0.60] | 0.58[0.56, 0.62] |
| **(c)** $B = 140$ | | | |
| Mistral | 0.59[0.55, 0.61] | 0.59[0.56, 0.64] | 0.59[0.55, 0.61] |
| Qwen | 0.61[0.59, 0.64] | 0.62[0.59, 0.65] | 0.52[0.49, 0.56] |
| Gemma | 0.53[0.50, 0.57] | 0.56[0.52, 0.58] | 0.55[0.51, 0.58] |
| Llama | 0.58[0.55, 0.60] | 0.59[0.55, 0.63] | 0.58[0.56, 0.61] |
| **(d)** $B = 160$ | | | |
| Mistral | 0.60[0.57, 0.63] | 0.58[0.56, 0.64] | 0.58[0.55, 0.61] |
| Qwen | 0.59[0.56, 0.62] | 0.60[0.58, 0.64] | 0.53[0.52, 0.56] |
| Gemma | 0.54[0.51, 0.59] | 0.57[0.53, 0.60] | 0.54[0.51, 0.59] |
| Llama | 0.58[0.54, 0.61] | 0.60[0.56, 0.62] | 0.59[0.57, 0.63] |

In fact, Figure 9 shows that the performance of AdaBoN improves as the per-prompt inference budget $B$ grows, although to a lesser extent than when $K$ increases.

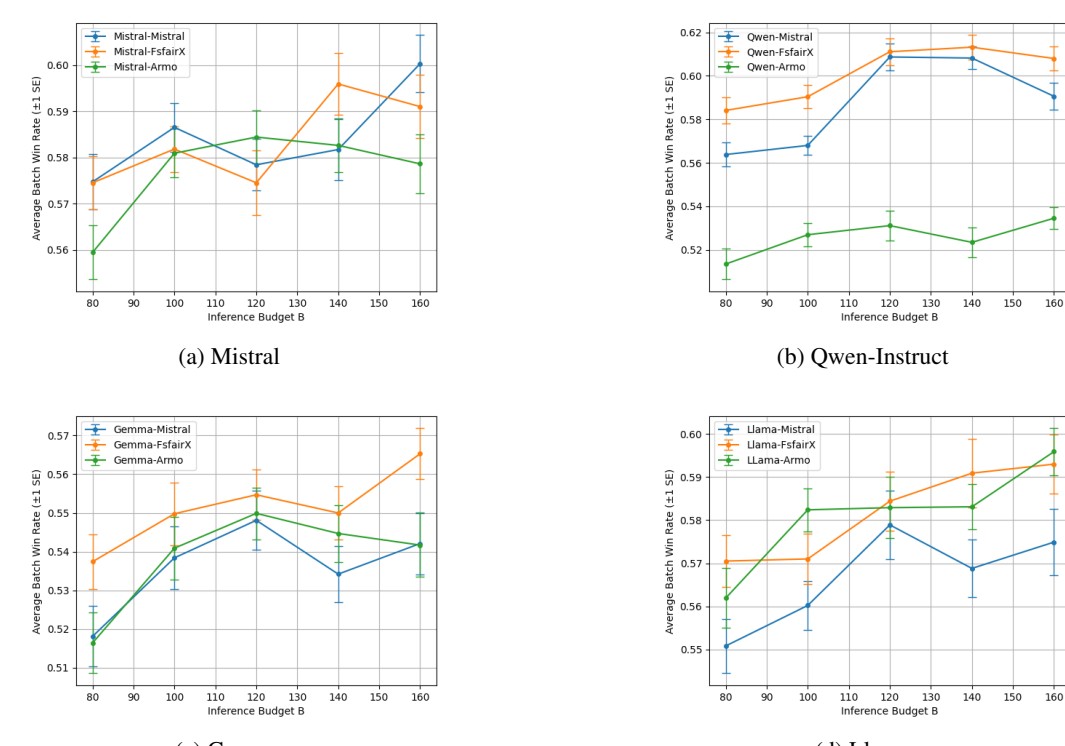

Figure 9: Average BWR ($\pm 1\,\mathrm{SE}$) as a function of $B \in \{80, 100, 120, 140, 160\}$ when $K = 5$ and $d = 0.75B$ on the **AlpacaEval dataset**. Generally, we observe an increase in BWR as $B$ increases, although the improvements are modest.

This is further substantiated by Table 13, where each cell is the median [Q1, Q3] of the differences $\mathrm{BWR}_{\mathcal{A}}(x_{1:K}^{(1)}, 160) - \mathrm{BWR}_{\mathcal{A}}(x_{1:K}^{(1)}, 80), \ldots, \mathrm{BWR}_{\mathcal{A}}(x_{1:K}^{(50)}, 160) - \mathrm{BWR}_{\mathcal{A}}(x_{1:K}^{(50)}, 80)$ across all 50 batches $x_{1:K}^{(1)}, \ldots, x_{1:K}^{(50)}$. Here, we observe strictly positive improvements in median per-batch BWR as $B$ increases from 80 to 160.

Table 13: Median [Q1, Q3] increase in BWR as budget increases from $B = 80$ to $B = 160$ on the **AlpacaEval dataset**, keeping $K = 5$ and $d = 0.75B$ fixed.

| LM | RM | | |
|---|---|---|---|
| | Mistral | FsfairX | Armo |
| Mistral | 0.022[-0.0088, 0.065] | 0.015[-0.015, 0.050] | 0.022[-0.024, 0.059] |
| Qwen | 0.040[-0.0037, 0.060] | 0.030[-0.015, 0.060] | 0.030[-0.014, 0.054] |
| Gemma | 0.028[-0.024, 0.069] | 0.028[-0.014, 0.066] | 0.030[-0.015, 0.049] |
| Llama | 0.020[-0.01, 0.060] | 0.025[0.0013, 0.045] | 0.033[0.0, 0.071] |

### K.2 VARYING BATCH SIZE $K$

Keeping $B = 120$ fixed, we vary $K$ with values in $\{3, 5, 10, 15, 20\}$. Since the result for $K = 5$ is presented in the main text, we only present the results for the other four choices of $K$. Tables 14 and 15 summarize these results and showcases that the performance of AdaBoN improves with the batch size $K$.

Table 14: Percent of batches with BWR $> 0.50$ for batch sizes $K \in \{3, 10, 15, 20\}$ on the **AlpacaEval dataset**. The per-prompt inference budget $B$ and exploration budget $d$ are fixed to 120 and $0.75B$ respectively.

| LM | RM | | |
|---|---|---|---|
| | Mistral | FsfairX | Armo |
| **(a)** $K = 3$ | | | |
| Mistral | 84% | 84% | 88% |
| Qwen | 96% | 100% | 70% |
| Gemma | 84% | 68% | 82% |
| Llama | 88% | 86% | 84% |
| **(b)** $K = 10$ | | | |
| Mistral | 98% | 100% | 100% |
| Qwen | 100% | 100% | 82% |
| Gemma | 86% | 88% | 80% |
| Llama | 100% | 100% | 100% |
| **(c)** $K = 15$ | | | |
| Mistral | 98% | 98% | 100% |
| Qwen | 100% | 100% | 88% |
| Gemma | 82% | 90% | 82% |
| Llama | 96% | 98% | 100% |
| **(d)** $K = 20$ | | | |
| Mistral | 100% | 100% | 100% |
| Qwen | 100% | 100% | 82% |
| Gemma | 88% | 86% | 88% |
| Llama | 100% | 100% | 100% |

Table 15: Median [Q1, Q3] BWRs for batch sizes $K \in \{3, 10, 15, 20\}$ for the **AlpacaEval dataset**, fixing budget $B = 120$, and exploration budget $d = 0.75B$

| LM | RM | | |
|---|---|---|---|
| | Mistral | FsfairX | Armo |
| **(a)** $K = 3$ | | | |
| Mistral | 0.55[0.52, 0.57] | 0.54[0.52, 0.56] | 0.54[0.52, 0.56] |
| Qwen | 0.56[0.54, 0.58] | 0.57[0.55, 0.58] | 0.53[0.50, 0.55] |
| Gemma | 0.54[0.51, 0.57] | 0.53[0.50, 0.56] | 0.54[0.52, 0.56] |
| Llama | 0.55[0.52, 0.58] | 0.55[0.52, 0.56] | 0.55[0.53, 0.58] |
| **(b)** $K = 10$ | | | |
| Mistral | 0.63[0.59, 0.67] | 0.61[0.58, 0.64] | 0.61[0.59, 0.66] |
| Qwen | 0.67[0.64, 0.70] | 0.67[0.64, 0.69] | 0.56[0.52, 0.61] |
| Gemma | 0.56[0.53, 0.60] | 0.56[0.53, 0.61] | 0.57[0.52, 0.60] |
| Llama | 0.62[0.59, 0.65] | 0.63[0.59, 0.66] | 0.65[0.62, 0.69] |
| **(c)** $K = 15$ | | | |
| Mistral | 0.66[0.61, 0.70] | 0.65[0.62, 0.70] | 0.65[0.62, 0.69] |
| Qwen | 0.71[0.67, 0.75] | 0.69[0.65, 0.72] | 0.57[0.54, 0.61] |
| Gemma | 0.57[0.53, 0.60] | 0.58[0.54, 0.62] | 0.57[0.52, 0.60] |
| Llama | 0.65[0.61, 0.68] | 0.66[0.60, 0.70] | 0.67[0.63, 0.70] |
| **(d)** $K = 20$ | | | |
| Mistral | 0.69[0.65, 0.72] | 0.66[0.63, 0.70] | 0.65[0.62, 0.70] |
| Qwen | 0.74[0.71, 0.77] | 0.72[0.70, 0.75] | 0.63[0.58, 0.68] |
| Gemma | 0.59[0.55, 0.62] | 0.59[0.53, 0.63] | 0.57[0.52, 0.64] |
| Llama | 0.67[0.64, 0.70] | 0.68[0.62, 0.72] | 0.69[0.65, 0.72] |

### K.3 IMPACT OF REWARD DISTRIBUTION ESTIMATOR

In this section, we present results for two other reward estimation procedures: Maximum Likelihood Estimation for the Gaussian and Gumbel distributions. We show that for all LM-RM pairs and datasets, these alternate reward estimations procedures perform worse than using Gaussian Kernel Density Estimation.

Table 16: Median BWRs for $K = 5$, $B = 120$, and $d = 0.75B$ for the three reward distribution estimations procedures we consider: Gaussian KDE (left), Gaussian MLE (middle), Skew-Normal MLE (right). We find that for the majority of LM-RM combinations, using the Gaussian KDE reward estimator results in the highest BWR, across all datasets.

| LM | RM | | |
|---|---|---|---|
| | Mistral | FsfairX | Armo |
| **AlpacaEval** | | | |
| Mistral | **0.58**, 0.56, 0.55 | 0.58, **0.61**, 0.56 | **0.59**, 0.57, 0.56 |
| Qwen | **0.60**, 0.53, 0.49 | **0.62**, 0.53, 0.49 | **0.54**, 0.51, 0.48 |
| Gemma | **0.56**, 0.52, 0.49 | **0.55**, 0.54, 0.53 | **0.56**, 0.54, 0.49 |
| Llama | 0.58, **0.59**, 0.55 | **0.59**, 0.58, 0.57 | **0.59**, 0.58, 0.56 |
| **HH-RLHF** | | | |
| Mistral | **0.58**, 0.56, 0.55 | **0.60**, 0.59, 0.56 | **0.55**, 0.53, 0.52 |
| Qwen | **0.55**, 0.53, 0.46 | **0.54**, 0.52, 0.47 | **0.53**, 0.52, 0.43 |
| Gemma | **0.54**, 0.51, 0.48 | **0.53**, 0.50, 0.48 | **0.55**, 0.52, 0.49 |
| Llama | **0.59**, 0.58, 0.52 | 0.57, 0.57, 0.53 | 0.57, **0.58**, 0.54 |
| **PKU-SafeRLHF** | | | |
| Mistral | **0.57**, 0.55, 0.58 | 0.57, **0.58**, 0.57 | 0.57, 0.57, 0.55 |
| Qwen | **0.54**, 0.53, 0.45 | **0.56**, 0.52, 0.46 | **0.49**, 0.47, 0.44 |
| Gemma | **0.53**, 0.52, 0.52 | **0.58**, 0.57, 0.54 | **0.57**, 0.56, 0.53 |
| Llama | **0.56**, 0.53, 0.51 | **0.59**, 0.58, 0.56 | **0.61**, 0.58, 0.57 |

