# OpenReview forum: "AdaBoN: Adaptive Best-of-$N$ Alignment"
_ICLR.cc/2026/Conference — Submitted to ICLR 2026_

### Official Review · Reviewer_LjAn · 2025-10-24

**Soundness:** 3
**Presentation:** 3
**Contribution:** 2
**Rating:** 6
**Confidence:** 4

**Summary:**

This paper proposes a method for adaptively allocating compute in parallel scaled inference-time alignment.  A popular way to scale inference-time compute is through the Best-of-N algorithm, where for each prompt, $N$ responses are generated and scored and the best is returned.  Typically the same $N$ is used for all prompts in a given batch, but this can be wasteful in settings where there is significant heterogeneity in difficulty of answering a question.  This paper proposes a way to adaptively select the number of responses to be used for each prompt in order to allocate a higher $N$ for responses that have a potentially higher reward.  The method they propose is to first sample some number of responses for each prompt and use these to form an estimate of the distribution of rewards using a kernel density estimator with a Gaussian kernel.  They then estimate the expected maximum reward using this density estimator and greedily assign allocation.  The authors conclude with an empirical study on several datasets and openweight models demonstrating improvement relative to the naive uniform allocation strategy.

**Strengths:**

This paper is timely in the sense that scaling inference time compute remains an important paradigm for improving LM performance and the question of how best to use such compute is essential.  The paper presents an interesting solution to the problem of how to allocate responses per prompt and demonstrates empirically that their solution nontriviallly separates from the naive uniform baseline.

**Weaknesses:**

I think it would be helpful for the paper to address the fact that in many important problems where best-of-N is used, the true reward we care about is binary, e.g., in math the ground truth reward is 1 if the correct answer is returned and zero otherwise.  Similar phenomena occur in rule-based rewards for safety and alignment purposes.  Including a discussion of this fact and how the paper relates to this setting seems important, especially as I suspect that the studied framework does not generalize to that setting.

Second, I think it would be useful to discuss the fact that learned rewards are often imperfect at the tails and thus best of N is susceptible to reward hacking.  To the extent that this method generates responses with higher estimated reward that are thus highly atypical, it would be good to discuss this issue.

Third, I am somewhat confused by the evaluation metric that the authors consider empirically, namely the batch win rate.  In most settings, I think people are more concerned with the per prompt win rate marginalized over prompts in a batch, not the aggregate reward over the batch.  This would manifest as $\frac 1 K \sum_{i = 1}^K \mathbb P\left(\max_{j \in [A_i]} R_{i,j} \geq \max_{j \in [B]} R_{i,j} \right)$ (in the simpler formulation that treats ties as wins).  I am concerned that the BWR as defined in the paper is susceptible to a single prompt having outsized influence and producing some very high rewards.  Could the authors please show some subset of their empirical results with this more standard notion of win rate to better demonstrate empirical efficacy?

Fourth, a minor point is that I think the authors are missing a reference to another paper that proposes an adaptive best-of-N approach that provides theoretical savings.  See e.g. Appendix D of *Self-Improvement in Language Models: The Sharpening Mechanism*.

**Questions:**

See weaknesses.

---

> ### Author Response · Authors · 2025-11-19
>
> Thanks for the review! We've attempted to address the most salient comments below. Please let us know if further questions arise.
>
> > I think it would be helpful for the paper to address the fact that in many important problems where best-of-N is used, the true reward we care about is binary, e.g., in math the ground truth reward is 1 if the correct answer is returned and zero otherwise. Similar phenomena occur in rule-based rewards for safety and alignment purposes. Including a discussion of this fact and how the paper relates to this setting seems important, especially as I suspect that the studied framework does not generalize to that setting.
>
> We appreciate the reviewer highlighting this important distinction. It is correct that our current method assumes a “continuous’’ reward model, making it less directly applicable to strictly binary rewards. However, the underlying framework and problem formulation are more general and can in principle extend to discrete or rule-based rewards. Exploring this extension is an active direction of our ongoing work, and we will add a discussion of this setting and its challenges in the final version.
>
> > Second, I think it would be useful to discuss the fact that learned rewards are often imperfect at the tails and thus best of N is susceptible to reward hacking. To the extent that this method generates responses with higher estimated reward that are thus highly atypical, it would be good to discuss this issue.
>
> We thank the reviewer for pointing out the very real concern of reward hacking. We will make a comment about this in the final version and also discuss the following recent NeurIPS paper: Inference-Time Reward Hacking in Large Language Models [1].
>
> [1] Khalaf, Hadi, et al. "Inference-Time Reward Hacking in Large Language Models." arXiv preprint arXiv:2506.19248 (2025).
>
> > Third, I am somewhat confused by the evaluation metric that the authors consider empirically, namely the batch win rate. In most settings, I think people are more concerned with the per prompt win rate marginalized over prompts in a batch, not the aggregate reward over the batch. This would manifest as (in the simpler formulation that treats ties as wins). I am concerned that the BWR as defined in the paper is susceptible to a single prompt having outsized influence and producing some very high rewards. Could the authors please show some subset of their empirical results with this more standard notion of win rate to better demonstrate empirical efficacy?
>
> In our allocation problem, the optimization objective is the sum of the rewards across the prompts in a batch. Because our method is not designed to provide a per-prompt guarantee, we defined the Batch Win Rate (BWR) in terms of the cumulative reward sum. We note that this cumulative reward objective has also been studied in prior work (e.g., Damani et al. 2024).
>
> That said, we agree that per-prompt win rate is a commonly used metric. In response to the reviewer’s suggestion, we are currently working on getting per-prompt win rates. We will upload a revised manuscript and make a comment once this data is ready.
>
> > Fourth, a minor point is that I think the authors are missing a reference to another paper that proposes an adaptive best-of-N approach that provides theoretical savings. See e.g. Appendix D of Self-Improvement in Language Models: The Sharpening Mechanism.
>
>  We thank the reviewer for pointing out this reference. We will make sure to include it in the final  version.

---

> > ### Author Response · Authors · 2025-11-20
> > **New Revision with Per-prompt Win Rates**
> >
> > Thanks again for the review! We have uploaded a revised version of the manuscript with the pre-prompt win rates you suggested in Appendix J Despite AdaBoN only optimizing for the cumulative sum of rewards (and hence the BWTR), we find that it's still competitive with the uniform allocation on a per-prompt basis (achieving win rates around 0.50-0.51) across all LLM-RM pairs for the AlpacaEval dataset.

---

> > > ### Comment · Reviewer_LjAn · 2025-11-20
> > >
> > > Thank you for your response and I appreciate the additional experiment.  Perhaps I was a bit unclear in my initial review, but I remain confused as to why the batch win rate is an interesting metric.  Other than the fact that it was studied in the prior work by Damani et al, why is it natural to consider this metric as opposed to the per-prompt win rate, which is an extremely common evaluation?
> > >
> > > Can you point to examples where we would care about the BWR as opposed to the more standard win rate?  I am struggling to think of concrete reasons where it would be acceptable to return one very high reward response to a single prompt and low reward responses to all other prompts in the batch, especially as in most settings (as I pointed out in my original review) the ground truth notion of reward is binary.  What are the examples you have in mind where the large reward range is meaningful outside of some estimated reward function?
> > >
> > > Moreover, if indeed it is the case that the standard win rate is the relevant metric as opposed to BWR, then the added empirical results, while appreciated, suggest to me that this approach is not doing anything to improve the relevant metric of win rate.  While not unexpected, as the method is tailored to BWR, it does make me wonder more about the motivation.

---

> > > > ### Author Response · Authors · 2025-11-22
> > > >
> > > > Thank you for the follow-up question! If we are not mistaken, it seems that you are ultimately questioning why the cumulative sum of rewards across the batch is an interesting objective to maximize (as opposed to the minimum reward across the batch). After all, maximizing the cumulative sum of rewards would maximize the BWR, while maximizing the minimum reward would maximize the more standard per-prompt win rate. We agree that for standard user-facing dialogue, the per-prompt win rate is the appropriate metric.
> > > >
> > > > That said, we believe that there are several applications where it is more meaningful to consider non-binary rewards and where the sum of rewards is a more natural objective. For example, BoN is often used to generate synthetic training data for weaker models. In this setting, it is more useful and expressive to have a continuous reward model, and one can argue that it is better to have a few very high-quality responses (i.e., a few high-reward samples) than to have a large number of marginally improved ones. The downstream utility of such data is often dominated by the best examples rather than the average quality across all prompts.
> > > >
> > > > Likewise, BoN is also commonly used in safety red-teaming, where the goal is to uncover harmful responses. Here, harmfulness is naturally graded rather than binary, and a small number of extremely harmful responses can be significantly more informative than many mildly harmful ones, as they reveal critical failure modes. In such cases, optimizing for the sum of rewards better captures the objective of surfacing extreme or high-impact behaviors.
> > > >
> > > > More generally, we believe that our objective of maximizing the sum is particularly appropriate when the outputs of the LLM are being used as data, rather than as direct responses to users.

---

> > > > > ### Comment · Reviewer_LjAn · 2025-11-26
> > > > >
> > > > > Sorry, perhaps I was unclear.  I am not asking anything about the minimum reward.  I am still confused about why we care about BWR as opposed to the more standard per-prompt win rate.  The authors' response appears more geared at the point that real-valued rewards (as opposed to binary rewards) can be natural in some particular settings, a point with which I agree.  Even in those settings, however, I am still struggling to understand why BWR is a natural metric.  In the case of synthetic training data, for example, I agree that it is often useful to have at least one correct response, but normally this would be true per prompt.  Why would it be the case that we wish to optimize for one very high reward response out of a large batch of prompts?
> > > > >
> > > > > I also find the safety example lacking as the BWR is not explicitly optimizing for one particular very high reward response, it is simply conflating that scenario with many mildly preferable responses.  If it is the case that we wish to optimize for one particular very high reward response, would not a more natural metric be something like average reward as opposed to something like win rate that is scale insensitive?

---

> ### Author Response · Authors · 2025-11-28
>
> > Even in those settings, however, I am still struggling to understand why BWR is a natural metric. In the case of synthetic training data, for example, I agree that it is often useful to have at least one correct response, but normally this would be true per prompt. Why would it be the case that we wish to optimize for one very high reward response out of a large batch of prompts?
>
> Thanks again for the follow-up! In practice, we have a limited inference budget, which constrains our ability to individually optimize for every prompt in our batch. With infinite compute, we agree with the reviewer that we would optimize per-prompt in our batch. However, when the budget is limited, we need to be more careful about how we distribute this compute across the prompts in the batch. In particular, under a budget constraint for synthetic data generation, we think it is reasonable to optimize a batch-level objective like the sum of per-prompt rewards. This balances improving many prompts moderately with improving some prompts substantially (and not necessarily just one very high-reward response).
>
> > I also find the safety example lacking as the BWR is not explicitly optimizing for one particular very high reward response, it is simply conflating that scenario with many mildly preferable responses. If it is the case that we wish to optimize for one particular very high reward response, would not a more natural metric be something like average reward as opposed to something like win rate that is scale insensitive?
>
> We would like to point out that we are optimizing for the sum of the rewards, and this does not necessarily mean we are optimizing for one particular very high-reward response -- the latter implies the former, but not the other way around. Moreover, the BWR is monotonically increasing with the sum (and average) of rewards. Thus, larger BWRs correlate with larger reward sums (or averages). That said, the reviewer is correct that the sum (or average) of the rewards is a reasonable metric. However, the outputs of reward models are often only meaningful comparatively, especially those trained on preference data via the Bradley–Terry model, since these often just output logits (which is true of the reward models we consider). This is especially true for alignment and preference modeling tasks, where the reward model serves only as a proxy for true unknown utility, and hence its outputs should be treated comparatively. That is why we opted to report the BWR, as it is scale-insensitive, as the reviewer noted.

---

### Official Review · Reviewer_K2hQ · 2025-10-31

**Soundness:** 2
**Presentation:** 1
**Contribution:** 2
**Rating:** 4
**Confidence:** 3

**Summary:**

Recent advances in test-time alignment methods, like Best-of-N sampling, enhance language models' behavior steering using reward models but can be computationally intensive when uniformly applied across prompts. This work introduces a prompt-adaptive strategy with a two-stage algorithm that efficiently allocates inference-time compute, outperforming uniform allocation and remaining competitive even with larger budgets.

**Strengths:**

1. The authors propose a simple yet effective two-stage Adaptive Best-of-N (AdaBoN) allocation scheme: in the first stage, a small exploration budget is used to estimate reward distributions for each prompt, and in the second stage, these estimates help compute the marginal value of additional samples, with a greedy algorithm assigning the remaining budget accordingly.

2. Two new evaluation metrics are introduced, termed the Batch Win Rate (BWR) and Expected Survival Time (EST), to assess the performance of the alignment strategy, which can help the community to assess the performance in similar setting.

3. Empirical findings reveal that reward distributions are mostly smooth and can be skewed by Kernel Density Estimation (KDE).

**Weaknesses:**

1. The proposed method may be impractical when dealing with batches containing prompts of varying difficulty levels, as simple prompts requiring short responses must wait for the completion of more complex prompts needing longer, reasoning-intensive responses, leading to inefficiency; a token-based allocation approach could be more practical.

2. The paper lacks a clear definition and discussion of variables V and Z in equation (2), which undermines its rigor.

3. The two-stage method can be costly, as it requires calling the language model twice in parallel, doubling the time cost compared to a fixed budget approach; a single-stage method to estimate prompt difficulty and allocate the budget initially might be more effective.

4. The current reward distribution is one-dimensional, focusing on a single property like helpfulness, which may be smooth; however, using a multi-property reward model could require a larger budget to estimate higher-dimensional distributions, potentially leading to the curse of dimensionality.

**Questions:**

See the weakness.

---

> ### Author Response · Authors · 2025-11-19
>
> Thanks for the review! We've attempted to address the most salient comments below. Please let us know if further questions arise.
>
> > The proposed method may be impractical when dealing with batches containing prompts of varying difficulty levels, as simple prompts requiring short responses must wait for the completion of more complex prompts needing longer, reasoning-intensive responses, leading to inefficiency; a token-based allocation approach could be more practical.
>
> Our method is in fact designed to handle batches containing prompts of varying difficulty. In the second stage of our approach (the greedy allocation stage), the remaining inference budget is allocated in proportion to the estimated difficulty of each prompt. That is, prompts with higher potential reward improvement from additional computation receive more samples. Importantly, prompt “difficulty’’ is not necessarily tied to the response length.
>
> That said, the reviewer raises a valid concern: because rewards are computed only on full responses, the method must wait for all completions within a batch. This can indeed introduce inefficiency when average response lengths vary significantly across prompts. The reviewer’s suggestion of a token-based allocation scheme is a promising direction, but implementing it would require a much more fine-grained reward model (e.g., process reward models capable of assigning token-level rewards). Unfortunately, such models are difficult to train reliably since they rely on high-quality, costly manual annotations. We will clarify these points and discuss this limitation in the revised version.
>
> > The paper lacks a clear definition and discussion of variables V and Z in equation (2), which undermines its rigor.
>
> Thank you for pointing this out. To clarify: in equation (2), the variable $V$ is \emph{defined} by the expression on the right-hand side, as indicated by the use of ``$:=$''. The variable $Z$ is introduced using the standard notation $Z \sim \hat{D}_i$, which specifies that $Z$ is a random variable drawn from the distribution $\hat{D}_i$.
>
> > The two-stage method can be costly, as it requires calling the language model twice in parallel, doubling the time cost compared to a fixed budget approach; a single-stage method to estimate prompt difficulty and allocate the budget initially might be more effective.
>
> A prompt-adaptive single-stage method would require training an auxiliary model to predict prompt difficulty, as in the approach of Damani et al. 2024. However, their results indicate that such methods do not scale well with increasing inference-time budget. In contrast, our two-stage method avoids the need to train an auxiliary predictor and instead uses reward samples to estimate prompt difficulty directly. This approach naturally scales with the inference budget: additional budget simply yields more accurate estimates of the reward distributions. We will clarify this tradeoff and the motivation for the two-stage design in the revised version.
>
> > The current reward distribution is one-dimensional, focusing on a single property like helpfulness, which may be smooth; however, using a multi-property reward model could require a larger budget to estimate higher-dimensional distributions, potentially leading to the curse of dimensionality.
>
>  Many extensions are possible and are beyond the scope of this work. We believe that developing a rigorous and effective solution for the single-dimensional case is a necessary precursor to addressing a multi-dimensional setting.

---

### Official Review · Reviewer_m76B · 2025-11-01

**Soundness:** 2
**Presentation:** 2
**Contribution:** 2
**Rating:** 4
**Confidence:** 4

**Summary:**

This paper proposes AdaBoN, a two-stage adaptive allocation strategy to improve the computational efficiency of Best-of-N (BoN) sampling. It addresses the key limitation of standard BoN, which inefficiently applies a fixed, uniform sampling budget to all prompts regardless of their alignment difficulty. AdaBoN's first stage uses a small exploration budget ($d$) to sample each prompt and model its reward distribution using Gaussian Kernel Density Estimation (KDE). The second stage uses these distribution estimates to calculate the expected marginal gain for each prompt, and then greedily allocates the remaining inference budget to the prompts most likely to benefit from additional sampling. This method is model-agnostic and requires no auxiliary model training. Experiments on the AlpacaEval, HH-RLHF, and PKU-SafeRLHF datasets show that AdaBoN consistently outperforms the uniform allocation baseline and remains competitive against uniform methods using 20% larger budgets

**Strengths:**

The paper's primary strength is its simple, practical, and well-motivated solution to the clear inefficiency of standard Best-of-N (BoN) sampling. The proposed AdaBoN method is an elegant two-stage algorithm that is model-agnostic, requires no auxiliary model training , minimizes latency through parallelizable calls , and needs minimal hyperparameter tuning . These claims are substantiated by empirical evaluation across 3 datasets and 12 LM-RM pairs. The results consistently show that AdaBoN outperforms the uniform allocation baseline and remains competitive even against a uniform baseline given a 20% larger inference budget, demonstrating clear computational savings.

**Weaknesses:**

1. **Lack of Comparison to Relevant Adaptive Methods**: Absence of any empirical comparison against other adaptive allocation methods. The paper's baseline is limited exclusively to "uniform allocation" (i.e., standard Best-of-N sampling). The authors themselves identify Damani et al. (2024) as "the most closely related work", as it addresses the exact same problem of inference budget allocation. The authors justify omitting this crucial comparison by citing the lack of a public implementation and the "computationally prohibitive" cost of reproducing the method. While these practical hurdles are understandable, it leaves the paper's core contribution unevaluated against any relevant competition. By only outperforming a simple, non-adaptive baseline, the paper demonstrates an improvement but fails to show that this specific approach is a state-of-the-art or even a competitive solution. The claims of efficiency and effectiveness would be substantially stronger if AdaBoN were benchmarked against at least one other adaptive method from the related work section.

2. **Limited Scope of Tasks and Reward Distributions**: The paper's empirical validation is confined to a single class of problems: open-ended alignment tasks (AlpacaEval, HH-RLHF, PKU-SafeRLHF) evaluated by real-valued reward models. This narrow scope makes it difficult to assess the generalizability of the proposed method. It is unclear how this method would perform on tasks with fundamentally different reward structures, such as math reasoning (e.g., GSM8K) or coding, where reward distributions are more likely to be sparse.

**Questions:**

1. Given that reproducing Damani et al. (2024) is "computationally prohibitive", could you provide a comparison against a simpler adaptive heuristic (e.g., a greedy allocator that uses the variance of the initial $d$ samples) to better situate your method's performance against something other than uniform allocation?

2. How robust is the Gaussian KDE estimator to the sparse or highly bimodal reward distributions found in tasks like math reasoning (e.g., GSM8K)? Does this estimation method risk poor allocation if the distribution assumption is violated?

---

> ### Author Response · Authors · 2025-11-19
>
> Thanks for the review! We've attempted to address the most salient comments below. Please let us know if further questions arise.
>
> > Given that reproducing Damani et al. (2024) is "computationally prohibitive", could you provide a comparison against a simpler adaptive heuristic (e.g., a greedy allocator that uses the variance of the initial  samples) to better situate your method's performance against something other than uniform allocation?
>
> Please see the general comment above. As suggested, we have uploaded a revised manuscript comparing AdaBoN to a simple variance-based adaptive allocation policy.
>
> > How robust is the Gaussian KDE estimator to the sparse or highly bimodal reward distributions found in tasks like math reasoning (e.g., GSM8K)? Does this estimation method risk poor allocation if the distribution assumption is violated?
>
> Our work focuses primarily on alignment tasks, where rewards tend to be relatively continuous because they are trained from preference data. In this setting, we evaluated several density estimators (see Appendix K.3) and found that Gaussian KDE performed best among the smooth estimators we considered. You are right that for sparse or discrete reward distributions, a Gaussian KDE is not the ideal choice. When the distribution deviates substantially from continuity assumptions, KDE can misrepresent the mass at isolated reward values and potentially lead to suboptimal allocation. While our current method is therefore not directly applicable to such discrete-reward settings, the broader allocation problem remains relevant. Extending our approach to handle discrete or mixed reward distributions is an active direction of our ongoing work.

---

### Official Review · Reviewer_UiAt · 2025-11-14

**Soundness:** 2
**Presentation:** 2
**Contribution:** 2
**Rating:** 4
**Confidence:** 3

**Summary:**

The paper studies how to allocate a fixed Best-of-N (BoN) sampling budget across a batch of prompts using a reward model. It proposes AdaBoN, a two-stage, prompt-adaptive allocation scheme: a small exploration budget is first used to estimate per-prompt reward distributions via Gaussian KDE, then the remaining budget is greedily assigned to prompts with the largest estimated marginal gain in reward. The method is model-agnostic, requires no extra training, and operates with only two LM calls per batch. Experiments on AlpacaEval, HH-RLHF, and PKU-SafeRLHF with multiple LM–RM pairs show that AdaBoN consistently improves over uniform BoN with the same budget and can match or beat uniform BoN that uses a considerably larger budget.

**Strengths:**

1. The method is simple, test-time only, and easy to plug into existing BoN pipelines without retraining or heavy tuning.
2. Empirical evaluation spans several datasets and LM–RM pairs, with ablations over batch size, budget, and exploration fraction, supporting the main claims.
3. The proposed BWR and EST metrics are well aligned with practice and make the compute–quality trade-offs interpretable for BoN-style alignment.

**Weaknesses:**

1. Comparisons are restricted to uniform BoN; there is no empirical baseline from other adaptive allocation or difficulty-aware methods discussed in related work.
2. The paper does not quantify wall-clock overhead or throughput, leaving the practical cost of KDE fitting and Monte Carlo estimation somewhat unclear at larger scales.
3. The method's practicality at scale remains unclear, as its two-stage design and synchronous batching may struggle with heterogeneous prompts, and the paper offers limited evidence on how well it handles such real-world variability.

**Questions:**

1. Can the authors include simple adaptive baselines (e.g., variance- or entropy-based allocators) to better contextualize AdaBoN beyond uniform BoN?

---

> ### Author Response · Authors · 2025-11-19
>
> Thanks for the review! We've attempted to address the most salient comments below. Please let us know if further questions arise.
>
>
> > Comparisons are restricted to uniform BoN; there is no empirical baseline from other adaptive allocation or difficulty-aware methods discussed in related work.
>
> Please see the general comment above. As suggested, we have uploaded a revised manuscript comparing AdaBoN to a simple variance-based adaptive allocation policy.
>
> > The paper does not quantify wall-clock overhead or throughput, leaving the practical cost of KDE fitting and Monte Carlo estimation somewhat unclear at larger scales.
>
>  We note that our method does not require fitting a full Gaussian KDE; rather, we only need to sample from it, as described in Lines 298-303. Nonetheless, we agree that reporting wall-clock overhead is important for assessing practical applicability. Below, we report the combined wall-clock time for the estimation and allocation procedures in AdaBoN for the Mistral-Mistral LLM-RM pair, averaged over the 50 batches of prompts from the AlpacaEval dataset.
>
> ```
> Average wallclock time of ABoN over 100 runs on each of the 50 batches: 0.082142 seconds
> Total runs: 5000
> Min time: 0.072613 seconds
> Max time: 0.135403 seconds
> Std time: 0.004159 seconds
> ```
>
> We observe that the average wall-clock time of just the estimation and allocation procedure in ABoN is only 0.08 seconds. This is negligible compared to the time required for the (batch) generation of 120 responses per prompt, which takes on the order of minutes. Similar wall-clock times are observed for the other LLM-RM combinations and datasets. We will include this data in the final version of the manuscript.
>
> >  The method's practicality at scale remains unclear, as its two-stage design and synchronous batching may struggle with heterogeneous prompts, and the paper offers limited evidence on how well it handles such real-world variability.
>
> We would like to point out that a prompt-adaptive single-stage method would require training an auxiliary model to predict prompt difficulty, as in the approach of Damani et al. 2024. However, their results indicate that such methods do not scale well with increasing inference-time budget. In contrast, our two-stage method avoids the need to train an auxiliary predictor and instead uses reward samples to estimate prompt difficulty directly. This approach naturally scales with the inference budget: additional budget simply yields more accurate estimates of the reward distributions. We will clarify this tradeoff and the motivation for the two-stage design in the revised version.
>
> That said, the reviewer raises a valid concern: because rewards are computed only on full responses, the method must wait for all completions within a batch. This can indeed introduce inefficiency when average response lengths vary significantly across prompts. A token-based allocation scheme is a promising direction, but implementing it would require a much more fine-grained reward model (e.g., process reward models capable of assigning token-level rewards). Unfortunately, such models are difficult to train reliably since they rely on high-quality, costly manual annotations. We will clarify these points and discuss this limitation in the revised version.

---

### Author Response · Authors · 2025-11-19
**Comparison against simple variance-based adaptive baseline**

We sincerely thank all reviewers for their thoughtful feedback. Several reviewers have proposed comparing our method to a simple variance-based adaptive baseline, which computes empirical variances of the reward for each prompt from the exploration stage, and allocates the remaining budget proportional to these variances. We think this is a great suggestion and have updated the manuscript with this baseline, which we denote as VarBoN. BWRs for AdaBoN vs VarBoN and VarBoN vs the Uniform Allocation can be found in Appendix I. From here, we find that the performance of VarBoN is comparable to (if not worse than) the uniform allocation and worse than AdaBoN across all LLM-RM pairs.

---

### Meta-Review · Area_Chair_ksxS · 2026-01-03

**Summary:**

All the reviews are not positive and they raised many issues related to significance, novelty and evaluation. The author response addressed some of concerns in some way. But overall, the quality of the paper seems below the bar of ICLR.

**Reviewer Concerns:**

Some concerns are not fully addressed:

Limited Comparative Baselines. This is mentioned by a few reviewers. While the authors have uploaded a revised manuscript comparing AdaBoN to a simple variance-based adaptive allocation policy, this might not be enough.

In the author response, the authors agreed with some limitations of this work such as the method's practicality at scale.

**Reviewer Scores:**

Most reviewers might not change the scores.

---

### Decision · Program_Chairs · 2026-01-26

Reject